# Cloning of the broadly effective wheat leaf rust resistance gene *Lr42* transferred from *Aegilops tauschii*

Guifang Lin [1], Hui Chen [2], Bin Tian [1,7], Sunish K. Sehgal[3], Lovepreet Singh[4], Jingzhong Xie [1,8], Nidhi Rawat[4], Philomin Juliana [5,9], Narinder Singh [1,10], Sandesh Shrestha [1], Duane L. Wilson[1], Hannah Shult[1], Hyeonju Lee[1], Adam William Schoen[4], Vijay K. Tiwari[4], Ravi P. Singh [5], Mary J. Guttieri[6], Harold N. Trick [1], Jesse Poland [1,11], Robert L. Bowden [6], Guihua Bai [2,6], Bikram Gill [1✉] & Sanzhen Liu [1✉]

The wheat wild relative *Aegilops tauschii* was previously used to transfer the *Lr42* leaf rust resistance gene into bread wheat. *Lr42* confers resistance at both seedling and adult stages, and it is broadly effective against all leaf rust races tested to date. *Lr42* has been used extensively in the CIMMYT international wheat breeding program with resulting cultivars deployed in several countries. Here, using a bulked segregant RNA-Seq (BSR-Seq) mapping strategy, we identify three candidate genes for *Lr42*. Overexpression of a nucleotide-binding site leucine-rich repeat (NLR) gene AET1Gv20040300 induces strong resistance to leaf rust in wheat and a mutation of the gene disrupted the resistance. The *Lr42* resistance allele is rare in *Ae. tauschii* and likely arose from ectopic recombination. Cloning of *Lr42* provides diagnostic markers and over 1000 CIMMYT wheat lines carrying *Lr42* have been developed documenting its widespread use and impact in crop improvement.

[1] Department of Plant Pathology, Kansas State University, Manhattan, KS 66506-5502, USA. [2] Department of Agronomy, Kansas State University, Manhattan, KS 66506-5502, USA. [3] Department of Agronomy, Horticulture and Plant Science, South Dakota State University, Brookings, SD 57006, USA. [4] Department of Plant Science and Landscape Architecture, University of Maryland, College Park, MD 20742, USA. [5] International Maize and Wheat Improvement Center (CIMMYT), 56237 Texcoco, Mexico. [6] Hard Winter Wheat Genetics Research Unit, USDA-ARS, Manhattan, KS 66506-5502, USA. [7] Present address: Syngenta Crop Protection, Research Triangle Park, Durham, NC 27709, USA. [8] Present address: State Key Laboratory of Plant Cell and Chromosome Engineering, Institute of Genetics and Developmental Biology, Innovation Academy for Seed Design, Chinese Academy of Sciences, 100101 Beijing, China. [9] Present address: Borlaug Institute for South Asia, Ludhiana, India. [10] Present address: Bayer R&D Services LLC, Kansas City, MO 64153, USA. [11] Present address: Center for Desert Agriculture, Biological and Environmental Science and Engineering Division (BESE), King Abdullah University of Science and Technology (KAUST), Thuwal, Saudi Arabia. ✉email: bsgill@ksu.edu; liu3zhen@ksu.edu

Leaf rust, caused by *Puccinia triticina* Erikss., is a prevalent disease limiting wheat production worldwide[1]. Yield reductions in susceptible wheat cultivars typically range from trace to 30% and may exceed 50%. The yield loss can be mitigated by the introduction of genetic resistance[2]. More than 70 leaf rust resistance genes have been characterized and named in wheat (https://shigen.nig.ac.jp/wheat/komugi/genes/symbolClassList.jsp). Six cloned race-specific site leaf rust resistance genes include *Lr10*[3], *Lr21*[4], *Lr1*[5], *Lr22a*[6], and *Lr13*[7] belonging the nucleotide-binding leucine-rich repeat (NLR) gene family, and *Lr14a* encoding a membrane-localized protein containing ankyrin repeats[8]. Two race-nonspecific genes have been cloned, including *Lr34* encoding an ABC transporter[9] and *Lr67* encoding a hexose transporter[10]. Cloned resistance genes may be useful in assembling transgenic multigene cassettes for developing strong and durable resistant varieties to combat fast-evolving fungal pathogens[11,12].

The leaf rust resistance gene *Lr42* was identified from accession TA2450 in a collection of the wheat wild relative *Aegilops tauschii* Coss. (DD, $2n = 14$), the diploid D-genome donor for hexaploid bread wheat (*Triticum aestivum* L., AABBDD, $2n = 42$)[13]. *Lr42* confers all-stage resistance to leaf rust. To date, the *Lr42* gene is effective against all reported races (isolates from 2020 and previous years) of the leaf rust fungus in the US[14–16]. The *Lr42* resistance locus was introduced to a bread wheat cultivar "Century" by direct crossing, followed by two backcrosses to Century, and was released in a germplasm line KS91WGRC11 in 1991[17]. KS91WGRC11 has been extensively used in CIMMYT wheat breeding programs and is represented as line "*Lr42*" in CIMMYT pedigrees[18,19]. Several KS91WGRC11-derived cultivars released by CIMMYT have outstanding yield potential. Field studies in Oklahoma showed that near-isogenic lines with *Lr42* introgressions had a 26% increase in yield and 9% increase in kernel weight, which was attributed to leaf rust resistance[20].

In this work, we undertake the cloning of the *Lr42* gene because of its extensive use in international breeding, broad effectiveness, possible association with yield-enhancing factors, the need for diagnostic markers, and the potential utility of the cloned gene in transgenic cassettes. The *Lr42* gene was previously mapped to the short arm of chromosome 1D (1DS) using hexaploid wheat mapping populations[13,21–23]. We employ BSR-Seq, a bulked segregant RNA sequencing method[24], to map *Lr42*. To eliminate interference from A-genome or B-genome homoeologous sequences from hexaploid parents, we construct two diploid mapping populations by crossing the resistant accession with susceptible accessions of *Ae. tauschii*. Another advantage of using diploid parents is that the phenotype of *Lr42* is stronger and easier to distinguish compared to the phenotype in hexaploid wheat. Fine-scale mapping identifies the candidate gene that is then confirmed to be *Lr42* by ectopic expression in a susceptible wheat line as well as by gene knockout mutagenesis. The results confirm that the candidate gene is required and sufficient for *Lr42*-mediated resistance.

## Results

### Genetic mapping identified candidate genes of *Lr42* on 1DS.
We developed the diploid *Ae. tauschii* populations for efficient genetic mapping by crossing the *Lr42* donor *Ae. tauschii* accession TA2450 with two leaf rust susceptible *Ae. tauschii* accessions, TA2433 (Fig. 1a) and TA10132 (Supplementary Fig. 1). $F_{2:3}$ individuals from both populations were phenotyped for leaf rust resistance at the seedling stage. We scored 100 $F_{2:3}$ families of the TA2450 x TA2433 population and identified 27 homozygous resistant (HR) and 21 homozygous susceptible (HS) $F_{2:3}$ families (Fig. 1b). Leaf tissues of these HR and HS $F_{2:3}$ family seedlings were separately pooled for BSR-Seq[24]. The BSR-Seq experiment mapped

*Lr42* at a locus close to the end of the short arm of chromosome 1D (1DS) (Fig. 1c), consistent with the mapping results from the other mapping population TA2450 x TA10132 (Supplementary Fig. 1), and from the previous *Lr42* mapping studies in hexaploid wheat[19,21–23]. The results indicated that the gene we mapped in the diploid populations is the same as the *Lr42* gene transferred to hexaploid wheat. Based on the BSR-Seq results, we identified single-nucleotide polymorphisms (SNPs) that were likely located near the *Lr42* gene and converted them to Kompetitive Allele Specific PCR (KASP) markers for genotyping $F_3$ and $F_4$ individuals from both mapping populations (Supplementary Data 1). The *Lr42* mapping interval was narrowed down to ~116 kb flanked by the two markers, pC43 at 8,655,291 bp and pC50 at 8,771,761 bp on 1DS, based on the *Ae. tauschii* reference genome Aet v4.0[25] (Fig. 1d). Note that pC43 is an effective co-dominant marker to select *Lr42*-carrying lines in both *Ae. tauschii* and bread wheat lines (Supplementary Data 2). The two markers are located within two genes, which flank three other genes including an intact NLR gene (AET1Gv20040300), an NLR fragment (AET1Gv20040500), and a protein kinase (AET1Gv20040200). Therefore, the three genes in the interval on the reference genome were prioritized as the candidate genes of *Lr42* (Fig. 1d).

### Overexpression supported the intact NLR as *Lr42*.
BSR-Seq provided not only genetic mapping information but also genome-wide gene expression data. Our BSR-Seq result showed that all three candidate genes in the *Lr42* mapping interval were expressed in uninfected seedling leaves in both resistant and susceptible *Ae. tauschii* lines. Sequence comparison of the candidate genes between the resistant and susceptible lines using RNA-Seq data revealed that only the candidate AET1Gv20040300 contained polymorphisms in transcribed regions (Supplementary Figs. 2 and 3). We then amplified the full-length coding region of AET1Gv20040300 from both the resistant donor, TA2450, and the susceptible accession TA10132, which confirmed polymorphisms between the two alleles and different lengths of encoded proteins (Fig. 2 and Supplementary Figs. 4 and 5). Both alleles with the maize ubiquitin promoter were separately transferred to a bread wheat cultivar "Bobwhite". In total, we obtained four independent positive transgenic $T_0$ lines carrying the *Lr42* resistance allele (*Lr42*) from TA2450 and two carrying the *lr42* susceptibility allele (*lr42*) from TA10132. $T_1$ and $T_2$ transgenic lines were evaluated for leaf rust resistance.

Bobwhite carries the leaf rust resistance gene *Lr26* that confers resistance to many leaf rust *P. triticina* races[26]. We screened three *P. triticina* races and found that Bobwhite was susceptible to race TFBJQ (Supplementary Table 1), which is virulent on *Lr26*. Infection with race TFBJQ revealed that all *Lr42* transgenic lines gained high resistance and two *lr42* transgenic lines were highly susceptible like Bobwhite (Fig. 2a). Gene expression analysis showed that the *Lr42* allele was expressed in all *Lr42* transgenic lines and *lr42* was expressed in both *lr42* transgenic lines (Fig. 2b). Note that expression of both *Lr42* and *lr42* was not detectable in Bobwhite. In summary, the transgenic experiment with the *Lr42* expression under the control of the ubiquitin promoter showed that expression of the single gene AET1Gv20040300 can induce the resistance response to the pathogen.

### Long-read local assembly shows only one intact NLR in the *Lr42* locus.
To understand the haplotype of the *Lr42* locus, publicly available whole genome sequencing (WGS) Illumina data were used to first examine polymorphisms between TA2450 (*Lr42* resistant line) and TA10132 (susceptible reference line)[27]. The result from Comparative Genomics Read Depth (CGRD) analysis using WGS data[28], which provides similarities and copy number variation of

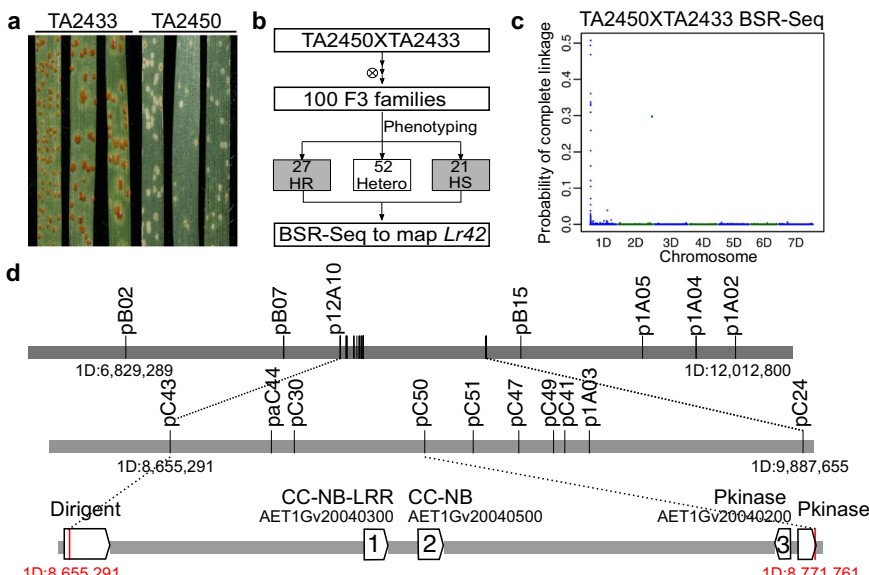

**Fig. 1 Genetic mapping of the *Lr42* gene. a** Phenotypes of *Ae. tauschii* accessions TA2433 (*lr42*) was susceptible (Infection Type = 33+) and TA2450 (*Lr42*) was hypersensitive fleck (Infection Type = ; to ;1−) at the seedling stage upon inoculation with race PNMRJ. **b** Genetic mapping *Lr42* genes via BSR-Seq. In total, we phenotyped 100 $F_{2:3}$ families (15 individuals for each family) from the cross TA2450 x TA2433 and identified 27 homozygous resistance (HR) families, 52 heterozygous (Hetero) families, and 21 homozygous susceptible (HS) families. Bulked segregant analysis via RNA-Seq (BSR-Seq) of 21 HR and 21 HS seedling pools was employed to map the *Lr42* gene. **c** For each variant identified from RNA-Seq, the probability of complete linkage between the variant and the *Lr42* gene was plotted versus the chromosomal position of the variant. **d** The first and second tracks display KASP markers developed from single-nucleotide polymorphisms identified by BSR-Seq. The markers pC43 and pC50 delineate the fine-mapping interval. On the third track, annotated genes overlapping with the interval are indicated with open boxes. Physical positions of markers are coordinates on the *Ae. tauschii* reference genome (Aet v4.0). Source data are provided as a Source Data file.

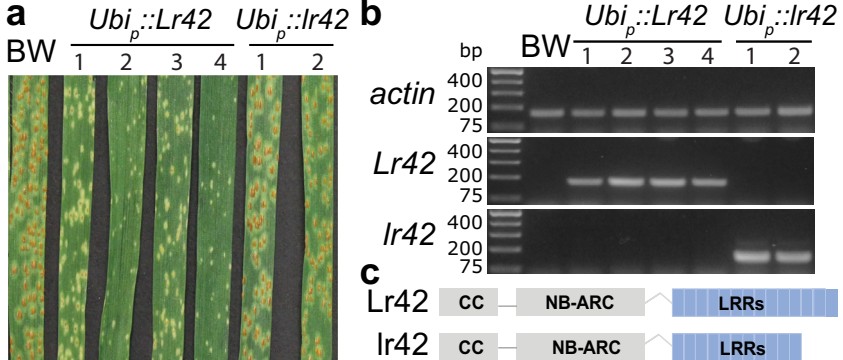

**Fig. 2 Constitutive expression of the *Lr42* candidate gene confers resistance. a** Phenotype of Bobwhite (BW, the variety used in transgenic experiments) was susceptible (Infection Type = 3), four independent transgenic lines of *Lr42* resistance allele (*Lr42*) showed hypersensitive fleck (Infection Type = ; to ;1−), and three independent transgenic lines of *Lr42* susceptibility allele (*lr42*) were susceptible (Infection Type = 3) upon inoculation with race TFBJG. Transgenic plants were tested in the $T_1$ generation. *Lr42* is from accession TA2450, and *lr42* is from accession TA10132. **b** RT-PCR of the *Lr42* gene of Bobwhite (BW) and transgenic plants from the $T_1$ generation. Resistant and susceptible alleles were amplified with primer sets Lr42-qRT-F5/R5 and lr42_1F/R. The amplicon lengths of *actin*, *Lr42* and *lr42* are 163, 171 and 142 bp, respectively. The experiment was repeated three times independently with consistent results. The *Lr42* expression result was confirmed in quantitative RT-PCR experiments. **c** The protein structures of the *Lr42* and *lr42* alleles. Note that *lr42* has fewer leucine-rich repeats (LRRs) compared with *Lr42*. Source data are provided as a Source Data file.

low-repetitive regions between the two genomes, revealed two relatively conserved segments (8.24–8.67 Mb and 8.70–8.82 Mb on 1D) and other divergent regions (Fig. 3a). We produced ~10x Nanopore long reads for the local assembly of the *Lr42* locus, resulting in a contig of 201,155 bp including the 116 kb *Lr42* mapping interval. Note this newly assembled sequence contains ~1% sequence errors. Sequence comparison showed that the *Lr42* locus contains all three candidate genes collinear with the genes in the reference genome (Fig. 3b). Consistently with RNA-Seq data, syntenic sequences of the protein kinase gene AET1Gv20040200 were almost perfectly aligned with 99.85% identity. The comparison

showed that the promoter region of AET1Gv20040300 is highly polymorphic. Besides AET1Gv20040500, we identified another NLR fragment homologous to AET1Gv20040300 (Fig. 3b). The expression of the fragment was not detected based on RNA-Seq data. Collectively, the sequence of the *Lr42* locus shows that the locus only contains a single intact NLR gene.

**Ectopic expression and knockout mutagenesis validate the NLR as *Lr42*.** To test the disease resistance of ectopic AET1Gv20040300 expression driven by the native promoter,

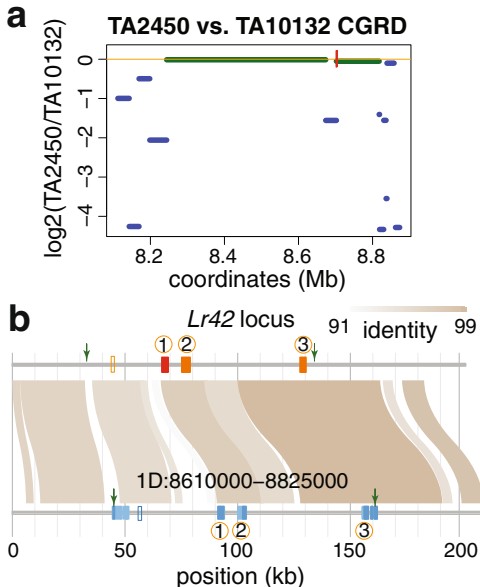

**Fig. 3 Sequence comparison of the *Lr42* locus with the reference. a** Read depth comparison between TA2450 and TA10132. *Y*-axis indicates the similarity between two genomes, green lines represent conserved regions and blue lines represent polymorphic or deleted regions in TA2450 relative to the reference genome (TA10132). **b** The sequence comparison between the assembled *Lr42* locus and a 1D region on the reference genome. On the *Lr42* locus, the red rectangle and orange rectangles stand for three genes syntenic with three annotated genes (1: AET1Gv20040300, an intact NLR; 2: AET1Gv20040500; 3: AET1Gv20040200) within the mapping interval in the reference genome. On the reference region, light blue and blue rectangles signify untranslated regions and coding regions of five annotated genes overlapping with the mapping interval, respectively. Blank rectangles represent the locations of NLR fragments homologous to AET1Gv20040300. Green arrows point to the flanking markers of the mapping interval.

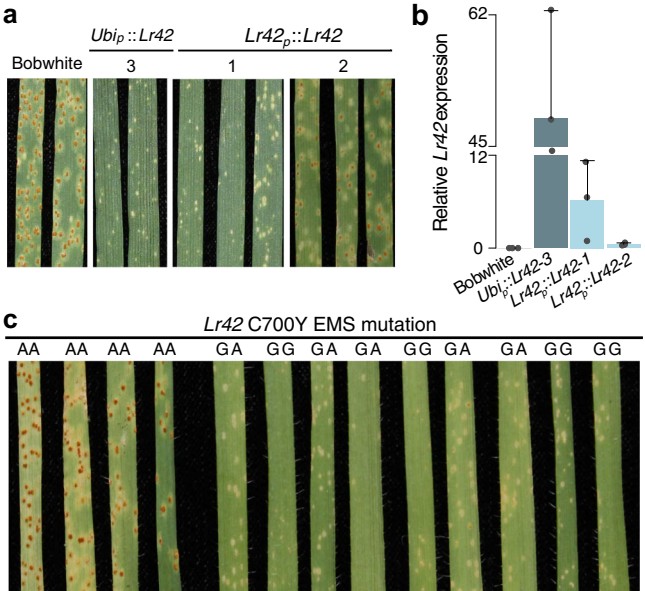

**Fig. 4 Ectopic *Lr42* expression and the EMS *Lr42* mutant. a** Phenotype of Bobwhite was susceptible ($N = 12$, Infection Type = 3), phenotype of *Ubi_p*::*Lr42* (*Lr42* driven by the maize ubiquitin promoter) $T_2$ transgenic plants (event 3) was hypersensitive fleck ($N = 12$, Infection Type = ;), and phenotype of two independent events of *Lr42_p*::*Lr42* (*Lr42* driven by the native promoter of *Lr42*) were resistant upon inoculation with race TFBJG. Infection types of *Lr42_p*::*Lr42-1* ($N = 3$) and *Lr42_p*::*Lr42-2* ($N = 9$) were ";" and 2−, respectively. Transgenic plants of both *Lr42_p*::*Lr42* events were in the $T_1$ generation, and *Ubi_p*::*Lr42* plants were from the $T_2$ generation. **b** qRT-PCR of the *Lr42* expression in Bobwhite and transgenic plants. *Lr42* expression of Bobwhite is undetectable. One plant for each biological replicate, and three biological replicates were used in gene expression analysis. Bar heights are means and error bars stand for standard deviations (SD). The quantification was repeated twice with consistent results. **c** Phenotype of M3 individuals from the TA2450 EMS mutant family that carried a G to A mutation at 2099 bp of the *Lr42* coding region, causing the C700Y amino acid substitution. The genotype (GG, GA, or AA) at the mutation site of each plant individual was listed above the leaf. Seedling plants were inoculated with race PNMRJ. Source data are provided as a Source Data file.

primers were designed to amplify the *Lr42* region, including the promoter, the *Lr42* gene, and the terminator. The promoter activity was validated with the GUS assay (Supplementary Fig. 6). Transformation of the gene with the native promoter in Bobwhite resulted in two events carrying the transgenic gene. The transgenic $T_1$ plants showed resistance to race TFBJQ (Fig. 4a). $T_1$ plants from one transgenic event (*Lr42_p*::*Lr42-1*) displayed a similar resistance level to the transgenic lines with constitutive expression driven by the maize ubiquitin promoter. $T_1$ plants from the other event (*Lr42_p*::*Lr42-2*) exhibited a weaker resistant phenotype. Expression quantification of *Lr42* via qRT-PCR indicated that a high level of rust resistance might require a certain threshold level of *Lr42* expression. At low levels of expression, resistance increased with the elevation of *Lr42* expression (Fig. 4b and Supplementary Fig. 7). Quantification of the genomic copy number of the *Lr42* transgene found that the lower *Lr42* expression transgenic line *Lr42_p*::*Lr42-2* contained a higher copy number of *Lr42* than *Lr42_p*::*Lr42-1* (Supplementary Figs. 7 and 8), possibly due to position effect and/or gene silencing associated with higher copy number of transgenes[29,30].

We then employed Virus Induced Gene Silencing (VIGS) to specifically knockdown expression of the AET1Gv20040300 gene in TA2450. Rust pustules were consistently observed on the leaves with reduced AET1Gv20040300 expression through VIGS using the construct containing a 201 bp fragment in the leucine-rich repeat (LRR) region (Supplementary Fig. 9). In contrast, the resistant phenotype was maintained on leaves through VIGS with no sequences targeting AET1Gv20040300 (Supplementary Fig. 9a).

The VIGS result confirmed that a high-level of rust resistance required a certain level of AET1Gv20040300 expression.

In addition, we screened 1320 M3 families of an Ethyl Methane Sulfonate (EMS) induced mutant population of the resistant accession TA2450[31] for their leaf rust responses. We identified one family showing the segregation for leaf rust resistance with 4 resistant and 9 susceptible individuals (Fig. 4c and Supplementary Fig. 10). The low positive mutation rate may be due to the loss of mutant alleles in the selected M3 families, a low number of seeds in many families, and weak phenotypes of missed mutants. Sequencing of the mutant found a G to A mutation in the LRR region, which causes a substitution from Cysteine to Tyrosine at the 700th amino acid (C700Y). Genotyping of all individuals ($N = 13$) in the family showed a co-segregation among genotypes and phenotypes, i.e., all homozygous mutants were susceptible and all others were resistant. Collectively, the results from the transgenic, VIGS, and EMS experiments consistently confirmed that the NLR gene AET1Gv20040300 is the *Lr42* gene.

**The *Lr42* resistance allele infrequently occurs in the *Ae. tauschii* collection.** The Wheat Genetics Resource Center (WGRC)

collected 549 *Ae. tauschii* accessions and has identified a minicore set of 40 accessions that capture >80% of genetic diversity of the whole set[32]. We examined the *Lr42* homologs from 35 minicore accessions, which include 24 accessions from Lineage 1 (L1, *Ae. tauschii* ssp. *tauschii*) and 11 from Lineage 2 (L2, *Ae. tauschii* ssp. *strangulata*)[32]. *Lr42* donor TA2450 belongs to L2. Expected bands were amplified from 8 out of 11 L2 accessions and 3 out of 24 L1 accessions (Supplementary Data 3). Of the bands amplified from 11 accessions, *Lr42* homologs from 10 accessions were successfully sequenced. We also extracted intact *Lr42* homologs from TA10132, the *Ae. tauschii* accession for the reference genome. TA10132, also known as AL8/78, is a leaf rust susceptible accession. Of all TA10132 *Lr42* homologs, the homolog with the highest similarity to *Lr42* and located in the *Lr42* mapping interval is deemed to be the allelic homolog of *Lr42* (*lr42-TA10132* or *lr42*). The *lr42-TA10132* allele was used in the transgenic experiment. Among all *Ae. tauschii Lr42* homologs, 10 homologs amplified from 10 *Ae. tauschii* minicore accessions are most similar to *Lr42*, supporting that these 10 homologs are also allelic to *Lr42* (Fig. 5a). The phylogenetic analysis indicated that the *Lr42* alleles are not completely separated in the two *Ae. tauschii* lineages (Fig. 5a). Sequences of the 11 *Lr42* allelic homologs, including *lr42-TA10132*, belonged to three major haplotypes I, II, and III, represented by *lr42-TA2376*, *lr42-TA1605*, and *lr42-TA2536*, respectively (Fig. 5b). Most sequences of the *Lr42* allele can be found from these three haplotypes except for a segment of ~140 bp in the LRR region, referred to as *Lr42*-unique-segment hereafter (Supplementary Fig. 4). Interestingly, *Lr42*-unique-segment can be identified with 98% identity in a non-allelic *Lr42* homolog from 1D subgenome of the Chinese Spring (CS) wheat reference genome (1D:7381846–738462, showing only 83.8% identity to *Lr42*) (Fig. 5b and Supplementary Figs. 11 and 12), implying that this unique sequence originated through either intragenic recombination or ectopic recombination. Beside the uniqueness of *Lr42*, we also observed conserved sequences at the end of the NB-ARC domain and at the beginning of LRR (Fig. 5c). A separate phylogenetic analysis using these domains (e.g., RX-CC, NB-ARC, and LRR) of the gene resulted in different phylogenetic relationships among these *Ae. tauschii* accessions, further supporting intragenic recombination occurred between *Lr42* haplotypes or ectopic recombination at some domains (Supplementary Fig. 4).

Among the accessions with the allelic *Lr42* homologs, two accessions TA2458 and TA2468 were seedling leaf rust resistant to race PNMRJ (Supplementary Fig. 13). Both *Lr42* haplotypes (*lr42-TA2458* and *lr42-TA2468*) were found in susceptible accessions (Fig. 5a), suggesting that the *Lr42* allelic homologs are not responsible for the leaf rust resistance in these two accessions. Indeed, TA2468 was known to carry *Lr21* that confers resistance to race PNMRJ (Supplementary Table 1)[4,33].

In the *Ae. tauschii* reference genome, the susceptibility *lr42* allele, four intact homologs, and four partial gene fragments were clustered within an 871 kb region (Fig. 5d). Interestingly, homologous sequences with plus and minus orientations were physically separated into two regions, and homologs with the same orientation are more similar. The organization of the gene cluster indicated that *Lr42* homologs likely expanded independently in the two separate regions. The *Lr42* homologous clusters were also identified in 1A, 1B, 1D subgenomes of the hexaploid wheat variety, CS, 1A and 1B chromosomes of tetraploid emmer wheat, 1A and 1B chromosomes of durum wheat, as well as 1H of diploid barley (Supplementary Data 4). Only two homologs were identified in Brachypodium, a more distantly related species (Fig. 5d). The results indicated that *Lr42* was derived from an ancient locus that has been maintained or expanded to result in a high copy number in barley and wheat species.

**Lr42 is a widely used source of effective resistance in wheat breeding programs**. Source germplasm lines KS91WGRC11 and KS93U50 carry the *Lr42* resistance alleles (Supplementary Fig. 14). These *Lr42* source lines carrying *Lr42* have been extensively used in the CIMMYT wheat breeding program. To identify which CIMMYT wheat lines containing the *Lr42* gene, both pedigree information and genotyping data via Genotyping-By-Sequencing (GBS) of 52,943 CIMMYT lines[34] were used. We identified 14 *Lr42*-specific GBS tags (Supplementary Table 2). Of 5121 genotyped CIMMYT lines with the *Lr42* introgression in the pedigree, 33.7% (1724/5121) were classified as *Lr42+* lines (Fig. 6a and Supplementary Data 5). In contrast, only 2% (928/47,822) of lines that were not expected to carry *Lr42* based on the pedigree were categorized as *Lr42+* lines. The 2% misclassified lines may reflect the false positive rate or could represent incorrect pedigrees or seed mixtures. In total, 2924 out of 5121 with an *Lr42* donor in the pedigree were categorized as without the *Lr42* segment (*Lr42−*) (Fig. 6a and Supplementary Data 5).

Some *Lr42+* and *Lr42−* wheat lines were phenotypically examined for leaf rust resistance and grain yield at CIMMYT[34]. Comparison between *Lr42+* and *Lr42−* wheat lines from the breeding population supported that the *Lr42* segment is highly associated with seedling resistance to the leaf rust race MBJ/SP[35], and moderately associated with resistance at the adult stage to leaf rust (Fig. 6b). Without leaf rust infection, grain yield traits of *Lr42+* and *Lr42−* lines were not significantly different, indicative of no significant yield boost or penalty directly imposed by the *Lr42* resistance segment from *Ae. tauschii* (Supplementary Table 3).

**Diagnostic markers for Lr42 genotyping**. We developed an effective co-dominant KASP marker pC43 that is located 46 kb from the *Lr42* gene for selection of *Lr42*-carrying lines in both *Ae. tauschii* and bread wheat lines. We have also designed and validated two markers, Lr42-pD1 and Lr42-pD2, on the *Lr42* gene to distinguish the presence or absence of the *Lr42* resistance allele in wheat (Supplementary Data 2, 6, and 7). These markers would be useful for precise marker-assisted selection of *Lr42* in wheat breeding programs.

## Discussion

We employed an efficient mapping strategy using diploid *Ae. tauschii* populations to clone the broadly effective leaf rust resistance gene *Lr42*. Cloning took advantage of the newly constructed *Ae. tauschii* reference genome[25], high-throughput sequencing technology, and the optimized genetic analysis strategy, BSR-Seq[24]. Using susceptible and resistant bulks, BSR-Seq enabled simultaneous discovery and genotyping of high-density SNPs to map the genomic region that contains *Lr42*. Further fine-mapping delimited the gene interval to ~116 kb interval and revealed an expressed candidate NLR gene, AET1Gv20040300, for *Lr42*. The causal gene was confirmed by gain-of-resistance via gene transfer to a susceptible hexaploid wheat cultivar as well as loss-of-resistance in an EMS mutant of the *Lr42 Ae. tauschii* diploid line.

The cloning of *Lr42* added a member to at least 12 known wheat rust resistance NLR genes (Supplementary Fig. 15)[3–6,36–42]. NLR functions as an intracellular sensor of pathogen signals and/or as an executor to induce localized cell death, the hypersensitive immune response. NLRs exerting both functions were recently referred to as singleton NLRs, such as *Mla*[43] and *Sr50*[44]. Some other NLRs function in a pair: sensor NLR recognizing the pathogen and helper (or executor) NLR initiating immune signaling. The paradigm of NLR networks consisting of a number of sensor NLRs and helper NLRs to modulate immune responses

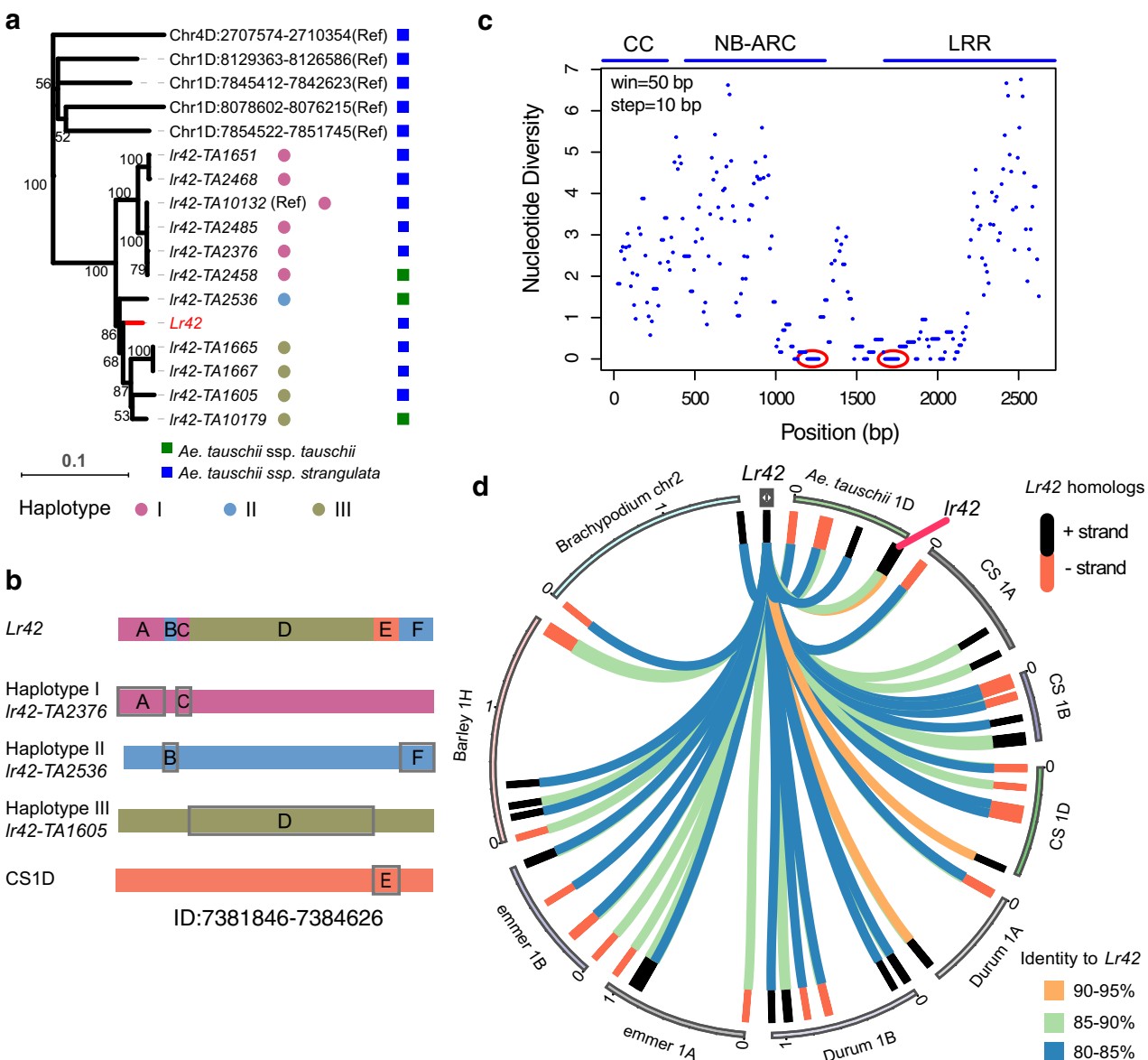

**Fig. 5 Homologs of *Lr42* in *Ae. tauschii* and closely related species. a** Phylogenetic tree of intact *Lr42* homologs from *Ae. tauschii*. The *Lr42* resistance allele is shown in red. The *lr42* reference allele is a susceptibility allele from the reference accession TA10132. The other ten alleles (signified by *lr42-*accession) were from the *Ae. tauschii* minicore set. Solid colored circles represent *Lr42* haplotypes. Squares indicate lineages of accessions. Bootstraps are labeled on the tree. **b** Haplotype analysis. Eleven *Lr42* alleles that do not include *Lr42* are clustered and grouped into three major haplotypes, represented by three alleles from TA2376, TA2536, and TA1605. Each sequence block (A to E) of *Lr42* indicates the best hit with at least 95% identity to the block with the same letter on three haplotypes and a sequence fragment from 1D subgenome of CS. **c** The nucleotide diversity of the 12 *Lr42* alleles in 50-bp windows scanned on the gene with a step size of 10 bp. Each dot represents a nucleotide diversity of a 50-bp window versus the middle position of the window. Conserved regions with very low diversity are highlighted by red ovals. **d** Circos view of *Lr42* homologs. *Lr42* clusters include *Lr42* homologs (at least 1 kb match and 79% identity) on *Ae. tauschii* 1D (7.84–8.71 Mb), bread wheat CS 1A (8.61–9.55 Mb), 1B (9.55–10.06 Mb), 1D (7.06–7.87 Mb), durum wheat 1A (8.46–9.21 Mb), 1B (8.11–9.21 Mb), Wild emmer 1A (10.41–11.48 Mb), 1B (11.95–12.87 Mb), Barley 1H (3.49–5.22 Mb), and Brachypodium chromosome 2 (38.09–39.65 Mb). The beginning of each cluster was adjusted to 0. The 1 s on the cluster track represent 1 Mb positions. The identity between each homolog and *Lr42* is color-coded. The red line points at the position of the *Lr42* susceptible allele on the *Ae. tauschii* reference genome. Source data are provided as a Source Data file.

was also proposed[45]. An NLR gene in monocots generally consists of an N-terminal coiled-coil (CC) domain, the central NB-ARC domain, and a C-terminal leucine-rich LRR domain. Recent protein structure studies of an Arabidopsis NLR gene product, ZAR1, revealed that a pentameric wheel-like NLR resistosome is assembled upon activation by the pathogen. The funnel-shaped structure formed from the N-terminal α helices at the CC domain is hypothesized to directly compromise plasma membrane

integrity and induce cell death[46,47]. Interestingly, a MADA motif (MADAxVSFxVxKLxxLLxxEx, where x represents non-conserved amino acids) conserved among helper NLRs and singleton NLRs but not sensor NLRs was identified on the CC domain[48]. *Lr42* has a typical NLR structure and contains a homologous domain "MAEAVVGQLVVTLGEALAKEA", which is most similar to the MADA motif among all known wheat rust resistance NLRs (Supplementary Table 4). This implies that *Lr42*

## a

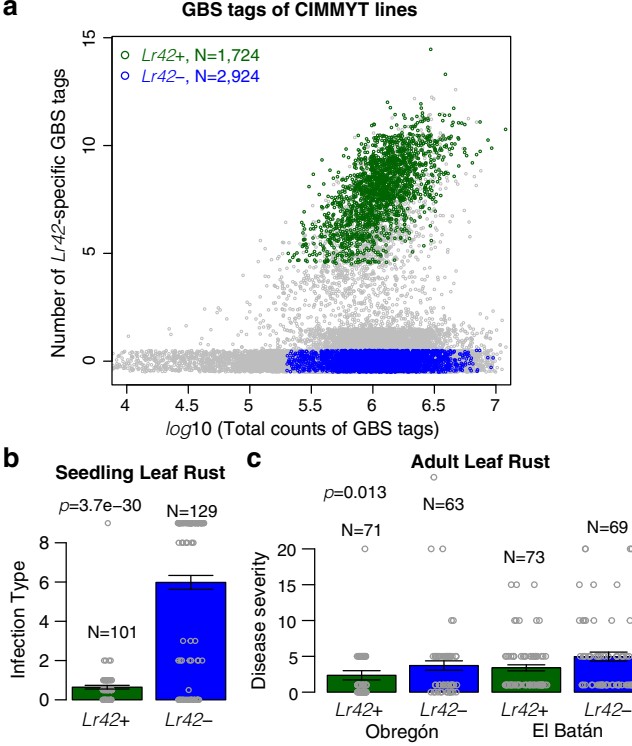

**GBS tags of CIMMYT lines**

*Lr42*+, N=1,724
*Lr42*−, N=2,924

Number of *Lr42*-specific GBS tags (y-axis)

*log*10 (Total counts of GBS tags) (x-axis)

## b

**Seedling Leaf Rust**

Infection Type

p=3.7e−30

N=129

N=101

*Lr42*+ *Lr42*−

## c

**Adult Leaf Rust**

Disease severity

p=0.013

N=71 N=63 N=73 N=69

*Lr42*+ *Lr42*− *Lr42*+ *Lr42*−
Obregón El Batán

**Fig. 6 Introgression of the *Lr42* segment in CIMMYT wheat lines. a** Each point represents a wheat line with the log10 of total count of GBS tags per line on the *x*-axis and the number of *Lr42*-specific GBS tags on the *y*-axis. Green and blue colors signify *Lr42*+ and *Lr42*− *Lr42*-introgressed lines, respectively. Other lines that are either not *Lr42*-introgressed lines or not confidently categorized into *Lr42*+ or *Lr42*− are gray colored. Numbers of *Lr42*-specific GBS tags were jittered with the factor 0.25 for better visualization of data density. **b**, **c** Using disease phenotypic data collected at CIMMYT, statistical comparisons of disease infection types between *Lr42*+ and *Lr42*− lines were performed for seedling stage leaf rust with two-side *t*-test and for adult stage resistance with ANOVA. Two field locations in Mexico for adult stage leaf rust phenotyping are labeled. The Stakman 0–4 scale was linearized to a 0–9 scale for seedlings (Methods). The percent disease severity on adult plant flag leaves was rated on the Cobb scale. Bars represent standard deviation of means of infection types or disease severity. Gray open dots represent single data points.

is more likely to be a singleton NLR or a helper NLR, not a sensor NLR.

*Lr42* is apparently a recently generated allele at an ancient locus. Homologs of *Lr42* were detected in the distant wheat relative Brachypodium (Fig. 5d), which diverged from the Triticeae (wheat, rye, barley) lineage 32–39 MYA[49]. Nevertheless, 34/35 samples in the *Ae. tauschii* minicore have been excluded to carry an *Lr42* resistance allele, suggesting that *Lr42* is not frequently present in the *Ae. tauschii* population and, likely, of recent origin. The variation in LRR repeat numbers among *Lr42* alleles indicated that unequal crossovers may have occurred within the LRR domain[50]. In addition, intragenic recombination as was documented for *Lr21*[33], or even ectopic recombination, may also have played a role in the origin of *Lr42* allele. Indeed, the unique LRR sequence of the *Lr42* allele can be identified in a non-allelic region in the subgenome 1D of CS, supporting the potential role of ectopic recombination in the origin of the *Lr42* resistance allele.

The phenotypic expression of resistance in *Lr42* lines depends on several factors. Although no leaf rust isolates have shown full virulence to *Lr42*, some isolates showed lower infection types than

others on KS93U50, an *Lr42* resistant selection from KS91WGRC11[22]. The resistance reaction of the diploid *Ae. tauschii* TA2450 donor accession is consistently very strong, ranging from a hypersensitive fleck (Infection Type (IT) = ;) to flecks with tiny pustules surrounded by necrosis (IT = ;1−) (Fig. 1a). However, the reaction of nontransgenic hexaploid *Lr42*-containing lines ranged from flecks and small pustules surrounded by necrosis (IT = ;1) to medium-sized pustules surrounded by chlorosis (IT = 2+)[13,22]. The reduced expression of introgressed resistance in hexaploid bread wheat compared to diploid donors is a frequently observed phenomenon[11]. However, the reaction of ubiquitin-driven transgenic hexaploid derivatives was very strong, ranging from a hypersensitive fleck (IT = ;) to flecks with tiny pustules (IT = ;1−) (Fig. 2a). The improved performance of the transgenic versus nontransgenic hexaploid lines may be due to the strong maize ubiquitin promoter that was used in the transgenics. The very strong resistance of the transgenic hexaploid *Lr42* lines bodes well for its utility in ubiquitin-driven transgenic cassettes. From the transgenic experiment with *Lr42* driven by the native promoter, we found one transgenic line with weak leaf rust resistance and low *Lr42* expression, which supported that the gene expression level is an important factor for *Lr42* resistance. In addition, plant age and/or environment may also influence *Lr42* resistance. Adult plants in the field showed much stronger *Lr42* resistance than greenhouse-grown seedlings[20].

The undefeated status of *Lr42* raised the possibility that it might be a more durable type of resistance gene. However, elucidation of the NLR structure of *Lr42* indicates that the mechanism of resistance is typical effector-triggered immunity (ETI). ETI is usually not durable because the rust pathogen can become virulent by loss of the corresponding avirulence factor (effector) that triggers the hypersensitive resistance response. It is possible that the effector gene conferring *Lr42* resistance is important for the fungus, which could explain why no virulent rust isolates have been identified. *Lr42* is currently deployed mainly in wheat lines from CIMMYT that contain combinations of durable adult plant resistance (APR) genes to leaf rust (Supplementary Table 5). This may have reduced the selection pressure on the pathogen population to overcome *Lr42*. The CIMMYT wheat breeding pipeline has many more *Lr42*-containing breeding lines in a background with high levels of APR to leaf rust (Supplementary Data 6). Effective gene stewardship will require breeders to release *Lr42* only in varieties with strong combinations of other leaf rust resistance genes.

Previous field trials showed that the *Lr42* introgression contributed to large increases in yield and kernel weight in Oklahoma[20]. We used GBS markers to classify 5121 CIMMYT breeding lines that had *Lr42* in the pedigree. Some of the advanced lines positive for *Lr42* were compared to their counterparts without *Lr42*. We were able to detect a very large effect of *Lr42* on leaf rust ratings at the seedling stage, but only a moderate effect on severity at the adult stage in the field probably because most CIMMYT lines also had a high level of APR that kept disease severities low (Fig. 6c). In a QTL analysis of highly resistant CIMMYT line Quaiu 3, Basnet et al. were able to separate the effect of *Lr42* from other resistance genes[19]. *Lr42* explained 32% of the phenotypic variation and limited disease severity in the field to a maximum of 40%. *Lr42* combined very well with *Lr46* and *QLr.tam-3D* to achieve near immunity to leaf rust in Quaiu 3[19]. We did not detect a direct or indirect impact of *Lr42* on yield and other grain quality traits, which is also probably due to a high level of APR in most CIMMYT lines.

KS91WGRC11 may be common in CIMMYT pedigrees because it contributes resistance to stem rust and stripe rust in addition to leaf rust. KS91WGRC11 carries the *SrTmp* stem rust resistance gene on chromosome 6DS from the Century parent[51].

We also documented a hidden introgression in the WGRC germplasm. Recently, a stripe rust resistant NLR gene *YrAS2388* originating from chromosome 4D of an *Ae. tauschii* accession was cloned and TA2450 was found to carry the resistance allele[42]. We amplified the *YrAS2388* gene from TA2450 and confirmed that the sequence is identical to the reference resistance allele reported. We found that KS91WGRC11 carries the *YrAS2388* resistance allele from TA2450, which implies that the *YrAS2388* resistance allele has been introduced to germplasm in CIMMYT and many other breeding programs. Given the limited backcrosses to Century, KS91WGRC11 is expected to harbor additional genomic segments from *Ae. tauschii* that might contribute valuable genetic diversity to future cultivars. Our results point to the need for in situ conservation of robust populations of native wild species for enhancing crop biodiversity so that alleles such as *Lr42* reported here can evolve and be conserved for future crop improvement.

## Methods

### Plant materials
*Ae. tauschii* accessions for genetic mapping and haplotype analysis are listed in Supplementary Data 3. *Ae. tauschii* ssp. *strangulata* accession TA2450 from Caspian Iran is the donor of the *Lr42* gene. Two highly susceptible accessions TA10132 (also known as AL8/78) and TA2433 were crossed with TA2450 and advanced to $F_{2:3}$, $F_{3:4}$, and $F_{4:5}$ populations by single seed descent.

### Leaf rust disease phenotyping
The leaf rust disease inoculation procedure followed the protocol developed by ref. [21] except plants were incubated in a growth chamber at 20 °C and a 16 h photoperiod. Briefly, for *Ae. tauschii* leaf rust phenotyping, two-leaf stage seedlings were inoculated with the leaf rust race PNMRJ. The virulence/avirulence phenotype of rust races is given in Supplementary Table 1. The infection type of plants was scored on the 0 to 4 Stakman scale at 10 days post inoculation (dpi) and confirmed at 14 dpi. The race nomenclature, differential sets, and Stakman infection types were described by refs. [1,14,52]. For transgenic wheat seedling leaf rust phenotyping, race TFBJG was used to inoculate seedlings at the two-leaf stage. TFBJG was used because it defeated *Lr26* in Bobwhite.

### BSR-Seq
Two $F_{2:3}$ populations (population 1: TA2450 x TA2433 and population 2: TA2450 x TA10132) were used for BSR-Seq analysis [https://schnablelab.plantgenomics.iastate.edu/software/BSR-Seq]. Fifteen seeds from each of $F_{2:3}$ families were inoculated and phenotyped. Of 100 TA2450 x TA2433 $F_{2:3}$ families, 27 were HR families of which all individuals were resistant to leaf rust, and 21 were HS families of which all individuals were susceptible. Equal amounts of leaf tissue were collected from each of 21 HR and 21 HS families, and HR and HS families were pooled separately. Of 101 TA2450 x TA10132 $F_3$ families, 36 were HR families and 9 were HS families. We selected 26 HR and all 9 HS families to collect HR and HS tissue pools.

RNA samples were extracted using the RNeasy Plant Mini Kit (Qiagen, Germany, Cat.# 74904) and with the $2 \times 101$ bp paired-end platform on an Illumina HiSeq2000 at the Genome Sequencing Facility at the Kansas University Medical Center. In total, ~180 million pairs of reads were generated. Raw reads were trimmed using Trimmomatic (version 0.32) [https://github.com/usadellab/Trimmomatic]. Trimmed reads were aligned to the *Ae. tauschii* reference genome (Aet v4.0, GCA_002575655.1)[25] by GSNAP (version 2018-03-25) [http://research-pub.gene.com/gmap][53] with the parameters of "-B 2 -N 1 -m 6 -i 2 -n 3 -Q". SNPs were discovered with GATK (version 3.3) [https://github.com/broadinstitute/gatk][54] with the UnifiedGenotyper module using the following parameters: "--heterozygosity 0.005 -stand_call_conf 30.0 -stand_emit_conf 20.0 -glm BOTH -U ALLOW_N_CIGAR_READS -ploidy 2". SNPs were filtered by the GATK module of SelectVariants with the following parameters: --restrictAllelesTo BIALLELIC --selectTypeToInclude SNP --select "AF >= 0.2 && QUAL >= 30.0 && DP >= 200 && DP < 10000". In total, 170,069 SNPs were identified for population 1 and 74,206 SNPs for population 2. For each population, a Bayesian-based approach was used to determine the probability of the complete linkage between each SNP and the causal gene[24].

### Fine mapping with KASP markers
SNPs having a high probability of the complete linkage with the causal gene were selected to convert to KASP assays. All KASP markers used for fine mapping were listed in Supplementary Data 1. The KASP experiment was run on the Applied Biosystems Real-Time PCR Instruments 7900 (Applied Biosystems, USA) using the KASP-TF Master Mix (LGC, Biosearch Technologies, UK, Cat.# KBS-1050-132) according to the manufacturer's instructions. To confirm the mapping interval, 68 $F_{2:3}$ families from the population 1 used for BSR-Seq were selected for genotyping. The 68 DNAs of pooled tissue samples from 12 $F_{2:3}$ individuals per family were genotyped with KASP markers p12A10,

p1A05, and p1A02. As a result, 11 $F_{2:3}$ recombinant families were identified. Analysis of genotypic data together with phenotypic data confirmed that the *Lr42* gene was located between the markers p12A10 and p1A05. To validate this interval, 6 of the 11 recombinant families were selected to genotype individual plants in each family with more KASP markers within the mapping interval.

To narrow down the mapping interval, we used $F_4$ plants from population 1. We first identified 9 $F_4$ families that were derived from the resistant $F_{2:3}$ individuals heterozygous for the *Lr42* in the mapping interval. In total, 891 $F_4$ individuals were phenotyped for rust resistance and genotyped with the markers p12A10 and p1A05, which identified 85 recombinants. Genotyping recombinants with additional markers identified nine $F_4$ individuals harboring the recombination between p12A10 and pC24. Further analysis of the $F_5$ progeny of these three $F_4$ individuals confirmed by the mapping interval between 8,655,291 bp and 8,830,775 bp on 1DS flanked by the markers pC43 and pC49.

We also analyzed 78 $F_2$ families of the population 2 and found four $F_2$ families with the recombination between p12A10 and p1A05. Luckily, one recombinant between the marker pC43 and the marker pC50 enabled us to locate the gene at a 116 kb interval between 8,655,291 bp and 8,771,761 bp.

### Cloning of full-length coding region of *Lr42* candidate gene
Total RNA was extracted from leaf tissues of resistant (TA2450) and susceptible (TA10132) accessions using TRIzol reagent (Invitrogen, USA, Cat.# 15596026) according to the manufacturer's instructions. After removing residual DNA with DNase I (Invitrogen, USA, Cat.# 18047019) treatment, 1 μg of total RNA was reverse-transcribed to cDNA using SuperScript® IV First-Strand Synthesis System (Invitrogen, USA, Cat.# 18091050) with an oligo(dT)$_{20}$ primer following the manufacturer's instructions. The full-length coding region of the *Lr42* candidate gene was amplified by PCR with the gene-specific primers AET300.2_CDS-F and AET300.2_CDS-R (Supplementary Data 8). The PCR product was cloned into the pCR-XL-2-TOPO vector (Invitrogen, USA). The inserted fragment in the construct was verified by sequencing using an ABI 3730 DNA analyzer (Applied Biosystems, USA).

### Construction of the plasmid of *Lr42* with the ubiquitin promoter
The full-length of *Lr42* coding regions flanked by a *Bam*HI restriction site was amplified via PCR using primer sets AET300.2_CDS-BamHIF and AET300.2_CDS-BamHIR (Supplementary Data 8). DNA fragments were ligated into a pAHC17 vector[55] at the *Bam*HI site. The expression constructs containing the full-length *Lr42* coding region under a maize ubiquitin promoter (*Ubi-1*) and a nopaline synthase terminator (tNOS) were used for generating transgenic plants.

### Assembly of the *Lr42* locus using Nanopore data
The public WGS Illumina data of TA10132 (the reference susceptible line, SRS7974112) and TA2450 (the *Lr42* parental resistant line, SRS7973948) were downloaded[27]. The data were used for genome comparison through CGRD [https://github.com/liu3zhenlab/CGRD][28], identifying conserved and variable regions on the *Ae. tauschii* reference genome between the two genomes.

To assemble the sequence of the *Lr42* locus, a low-depth (~10x) WGS Nanopore long reads of TA2450 were generated. Briefly, genomic DNAs were isolated from 12-day above-ground seedling tissues using a CTAB method[56]. A total amount of 2 μg TA2450 genomic DNA was used for the Oxford Nanopore library preparation. DNA was subjected to size selection using the BluePippin system (Sage Science, USA). The sequencing library was made using the ligation sequencing kit SQK-LSK109 (ONT, UK) and sequenced on a Nanopore PromethION sequencer (ONT, UK) at Wuhan Grandomics Biosciences co., ltd. The basecaller Guppy (version 4.2.2, Oxford Nanopore) [https://community.nanoporetech.com] was used to convert FAST5 raw data to FASTQ data with default parameters. WGS Nanopore reads were then aligned to the conserved regions close to the *Lr42* gene and the full-length *Lr42* gene sequence with minimap2 (2.21-r1071) [https://github.com/lh3/minimap2][57]. Reads with confident alignments are considered from the *Lr42* locus. Two sets of criteria were separately used to filter alignments to obtain confident alignments: (1) at least 8 kb matched sequence with at least 84% identity and less than 95% overhangs; (2) at least 12,000 kb matched sequence with at least 80% identity and less than 95% overhangs. Reads passing each set of alignment criteria were then separately assembled using flye (2.6) with the same parameters: "--min-overlap 1000 --asm-coverage 15 --genome-size 0.5 m --iterations 2" [https://github.com/fenderglass/Flye][58], resulting in two sets of contigs. Finally, both sets of contigs were manually checked and merged into a final assembly of the *Lr42* locus.

### Cloning the *Lr42* region including the promoter and terminator
The sequence data of the final assembly of the *Lr42* locus and WGS Illumina data were used to design primers for amplifying the *Lr42* region with the promoter and the terminator. The *Lr42* promoter region was predicted by the Softberry TSSP program (www.softberry.com). The *Lr42* sequence with the promoter and the terminator was covered by two *Lr42* fragments amplified with primer pairs of r42P_821F_F1/r42P_3725R and LR42_H1F/r42P_7872R_F1 (Supplementary Data 8). The two *Lr42* fragments have a 133 bp overlap that provided homologous recombination sequences for the DNA fragment assembly. Primers r42P_821F_F1 and

r42P_7872R_F1 (Supplementary Data 8) contained the *Eco*RI restriction enzyme site and the homologous sequence of the target vector pCR-Blunt (Invitrogen, USA). The *Eco*RI linearized pCR-Blunt vector and the two *Lr42* fragments were fused using the enzyme premix in ig-Fusion™ cloning kit (Intact Genomic, USA, Cat.# 4111). The expression construct was validated by Sanger Sequencing (Genewiz, USA) and used for generating transgenic plants.

**Activity assay of the *Lr42* native promoter.** The 2534 bp promoter fragment (containing TATA box) was amplified by PCR using a primer set Lr42p_BsaI_F containing a BsaI site and Lr42p_BamHI_R containing a BamHI site (Supplementary Data 8). The amplified fragment was digested with BsaI and BamHI, followed by ligation into BsaI/BamHI-digested vector pBI21 (Clontech, USA). This resulted in the *Lr42p*::*GUS* vector. The construct was verified by sequencing. *Agrobacterium tumefaciens* strain LBA4404 transformed with *Lr42p*::*GUS* plasmid was used to infiltrate wheat leaves as described by ref. [59]. GUS histochemical analyses were performed at 48 h after agroinfiltration according to ref. [60]. GUS expression driven by the *Lr42* promoter was detected by RT-PCR using a primer set GUS_RT_F and GUS_RT_R (Supplementary Data 8) as described by ref. [60]. The 18S rRNA gene was used as an internal control.

**Transgenic plants.** Immature embryos were isolated from a spring wheat (*Triticum aestivum* L.) cv. Bobwhite grown in a controlled environment with a 16-h photoperiod, and the day/night temperatures at 20/18 °C. The expression constructs and the pAHC20 vector[55] containing the *bar* gene were co-bombarded with 1:1 ratio into selected embryogenic calli. A biolistic approach using a particle inflow gun and following tissue culture protocols were performed for transformation[61,62]. Recovered plants in soil were screened for herbicide resistance by brushing a 0.2% v/v Liberty (glufosinate) solution (Bayer CropScience, USA) on leaves. The putative herbicide-resistant plants with an absence of necrosis after 5 days of Liberty application were analyzed by PCR for the presence of the gene of interest using primers Ubi-F and Seq2R (Supplementary Data 8) for *Ubip*::*Lr42* transgenic plants and Lr42_difBW_2F and Lr42-qRT_R6 (Supplementary Data 8) for *Lr42p*::*Lr42* transgenic plants. The transgenic plants were grown for leaf rust bioassays.

**EMS mutagenesis and screening.** The method of TA2450 EMS mutagenesis treatment was described by ref. [31]. In brief, 5300 seeds of TA2450 accession were soaked in 0.6% Ethyl methanesulfonate (EMS, Sigma-Aldrich, USA, Cat.# M0880-25G) for 8 h and then transplanted. The EMS-treated seeds and the plants grown from the EMS-treated seeds were in M0 generation, and M0 plants were self-pollinated to derive M1 seeds. The M1 seeds from a single M0 plant were collected as an M1 family. A single seed from each M1 family was randomly selected to generate M2 seeds, and the same procedure to generate M3 families. M3 seeds (*n* = 1320) were grown and inoculated with leaf rust race PNMRJ. Of the 1320 M3 families, 901 families had more than 16 seeds per family.

**Virus induced gene silencing of *Lr42*.** A Barley Stripe Mosaic Virus (BSMV)-based system developed by Yuan et al. was used for VIGS of *Lr42*[63]. BSMV vectors were shared by Lucy Stewart, USDA-ARS, Fort Detrick, MD. The primers (Lr42_BVIGS_P3_F1/R1, Supplementary Data 8), fused with Ligation Independent Cloning (LIC) sites, were designed to specifically amplify a 201 bp sequence of the *Lr42* allele. The PCR amplicon was cloned into a BSMVγ-LIC vector to generate BSMVγ-Lr42 construct[63]. BSMVγ-Lr42 was transformed into *Agrobacterium* GV3101 via electroporation and grown on LB plates containing 25 μg/ml rifampicin (Sigma-Aldrich, USA, Cat.# R7382-5G) and 50 μg/ml Kanamycin (Fisher Scientific, USA, Cat.# BP960-5). BSMVα-LIC, BSMVγ-Lr42, BSMVα and BSMVβ *Agrobacterium* cultures were grown separately overnight at 28 °C. Cultures were resuspended to OD$_{600}$ of ~0.7 in the infiltration buffer (10 mM MgCl$_2$ (Sigma-Aldrich, USA, Cat.# M2393-100G), 10 mM 2-(N-morpholino)-ethanesulfonic acid (MES) (pH 5.2) (Sigma-Aldrich, USA, Cat.# M2933-100G), and 0.1 mM acetosyringone (Sigma-Aldrich, USA, Cat.# D134406-5G) and incubated at room temperature for 4 h. Equal volumes of BSMVα, BSMVβ, and BSMVγ-Lr42 (or BSMVγ-LIC) cultures were mixed and infiltrated into three to four leaves of *Nicotiana benthamiana* plants using 1 ml needleless syringe. Infiltrated plants were maintained in the growth chamber. After 7 days post infiltration, 1 g of infiltrated leaf tissue was ground in 3 ml of ice cold 1x PBS containing 1% celite (Sigma-Aldrich, USA, Cat.# 20199-U) using a mortar and pestle. The sap containing viral particles were rub inoculated onto two-leaf stage seedlings of *Ae. tauschii* TA2450. After 2 weeks of viral inoculations, plants were infected with race PNMRJ as described above. At 14 days post rust inoculations, disease scores were recorded, and leaf samples were collected for gene expression analysis of *Lr42*.

**Gene expression analysis of *Lr42* in VIGS samples.** Total RNA was extracted using a Direct-Zol kit as per manufacturer's recommendation (Zymo Research, USA, Cat.# R2051). After DNase treatment with DNase I (New England Biolabs, USA, Cat.# M0303S), cDNA synthesis was performed using AzuraQuant cDNA synthesis kit (Azura genomics, USA, Cat.# AZ-1995). *Lr42* expression was monitored by performing qRT-PCR using the gene-specific primers amplifying the sequence outside the targeted region (LR42_qPCR_F1/R1, Supplementary Data 8). qRT-PCR was performed in 10 μl reaction volume containing 2 μl of 3X diluted

cDNA, 5 μl of 2X AzuraView GreenFast qPCR Blue Mix LR (Azura genomics, USA, Cat.# AZ-2301) and 400 nM of each of forward and reverse primers. Thermal cycler profile included 95 °C for 2 min and 40 cycles of 95 °C for 5 s and 60 °C for 40 s. Reactions were performed with three technical replicates. The *RL1* gene amplified by RLI_F1/R1 (Supplementary Data 8) was used as the reference gene for normalizing gene expression data[64]. Primer efficiencies for the target and reference genes were in the range of 100–110%, therefore, gene expression data were analyzed using the ΔΔCT method[28,65]. Three biological replicates were used for expression analysis.

**_Lr42_-specific GBS tags and identification of _Lr42+_ and _Lr42−_ CIMMYT wheat lines.** Both GBS data of *Ae. tauschii* accessions[32] and GBS data of CIMMYT lines[34] were used to identify *Lr42*-specific GBS tags that are associated with the *Lr42* segment from the *Ae. tauschii* donor TA2450. All GBS tags of TA2450 were aligned to the *Ae. tauschii* reference genome (v4.0)[25]. GBS tags that are located at the *Lr42* locus (~1 Mb upstream and downstream of the gene) and detected in less than 100 *Ae. tauschii* lines out of all *Ae. tauschii* collections at WGRC were considered to be associated with the *Lr42* segment. From the CIMMYT pedigree, 5121 CIMMYT lines were involved in the introgression of the *Lr42* segment from TA2450. Given missing data of GBS tags, we expect that each GBS tag that is specifically associated with the *Lr42* segment should be detected in less than 5000 lines. With that consideration, we obtained 14 *Lr42*-specific GBS tags (Supplementary Table 2), which were used to identify *Lr42+* and *Lr42−* wheat lines.

From the CIMMYT pedigrees, 5121 CIMMYT lines that were GBS genotyped could have the introgression of the *Lr42* segment from TA2450. The wheat lines carrying at least five *Lr42*-specific GBS tags were categorized as *Lr42+*, the lines harboring the *Lr42* segment. The wheat lines with no *Lr42*-specific GBS tags detected but with at least 0.2 million total GBS tags were categorized as *Lr42-*, the lines without the *Lr42* segment. All other lines were not classified.

**Phenotypic comparison between *Lr42+* and *Lr42−* CIMMYT lines.** Seedling plant responses of CIMMYT lines to leaf rust race MBJ/SP[35] were obtained using the original disease rating scale of 0–4 and converted to a 0–9 scale for the purpose of quantitative comparison using the conversion formula described in ref. [66]. The adult plant scoring was conducted using severity (0–100%, modified Cobb Scale). Seedling leaf rust responses were phenotyped in CIMMYT's greenhouses in El Batán and adult plant leaf rust responses were phenotyped in field trials at two locations, Ciudad Obregón and El Batán, in Mexico. Analysis of variance was performed to test the differential adult plant responses to leaf rust in two locations. T-tests were performed on seedling rust infection types and grain yield related traits, such as test weight and thousand kernel weight, evaluated as described by ref. [34].

**Haplotype analysis.** Genomic DNAs of leaf tissues from 35 *Ae. tauschii* accessions in the minicore collection from WGRC were extracted using 2% cetyl-trimethylammonium bromide (Sigma-Aldrich, USA, Cat.# H6269-250G)[67]. DNAs were used to survey sequences of *Lr42* haplotypes. *Lr42* alleles/homologs were amplified with the primers Lr42_H1F and Lr42_H1R (Supplementary Data 8) using Q5® High-Fidelity DNA Polymerase (New England Biolabs, USA, Cat.# M0491L) with High GC Enhancer. The PCR thermocycling conditions were initial denature at 98 °C for 3 min, 33 cycles of 98 °C for 8 s, 63 °C for 30 s, and 72 °C for 3 min, followed by a final extension at 72 °C for 5 min. PCR products were purified by using QIAquick Gel Extraction Kit (Qiagen, Germany, Cat.# 28706), and sequenced by Genewiz Sanger sequencing service. Sequencing reads were de novo assembled using Geneious software (version 8.1.7) [https://www.geneious.com]. The command cd-hit-est from the software CD-HIT (4.8.1) [https://github.com/weizhongli/cdhit] was used to cluster *Lr42* allelic homologs with default parameters[68]. The allele selected by cd-hit-est to represent each cluster was considered to be the haplotype sequence.

**Phylogenetic analysis.** ClustalW in the Geneious (version 8.1.7) was used for multiple alignment and phylogenetic construction. Multiple alignments were performed using the default setting. Phylogenetic trees were built with the Juke-Cantor model and the Neighbor-joining method. Trees were exported as Newick formatted flat files that were then uploaded to iTOL for plotting[69].

**Nucleotide diversity.** Nucleotide diversity of the 12 *Lr42* alleles was calculated by an R package, PopGenome [https://cran.r-project.org/web/packages/PopGenome][70]. Nucleotide diversity was calculated for windows with 50 bp and slided by the step of 10 bp.

**Identification of clusters of *Lr42* homologs.** Clusters of *Lr42* homologs were identified with BLAST (version 2.2.30+) [https://blast.ncbi.nlm.nih.gov][71]. First, the *Lr42* resistance allele was aligned to the genomes of *Brachypodium* (GCF_000005505.3 Brachypodium distachyon v3.0)[49], Barley (GCA_901482405.1_ Morex_v1.0)[72], *Triticum dicoccoides* wild emmer (GCA_002162155.2 WEW v2.0)[73], *Triticum turgidum* subsp. durum (GCA_900231445.1 Svevo.v1)[74], *Ae.*

*tauschii* (Aet v4.0)[25], and *T. aestivum* cv. CS (iwgsc_refseqv1.0)[75]. Homologs were identified if an alignment had the E-value smaller than 1e−100 and the matched length of the query (*Lr42*) was longer than 1 kb. Second, a chromosome interval smaller than 2 Mb with at least 2 homologs was identified as a *Lr42* cluster. Alignments of the *Lr42* resistant allele and homologs in each cluster were plotted using Circos [http://circos.ca][76].

**RNA extraction and RT-PCR of transgenic plants**. Leaf tissue from *Ae. tauschii* and transgenic wheat were collected, and RNA was extracted using the RNeasy Plant Mini Kit (Qiagen, Germany, Cat.# 74904). cDNA was synthesized with Verso cDNA Synthesis Kit (Thermo Scientific, USA, Cat.# AB1453A). The cDNA input for each sample was normalized by the housekeeping gene *actin* amplified with primers actin_F1 and actin_R1 (Supplementary Data 8) for 25 cycles. The *Lr42* resistant and susceptible alleles were amplified with primers Lr42-qRT-F5/R5 and lr42_1F/R (Supplementary Data 8) for 28 cycles. The OneTaq 2x Master Mix (New England Biolabs, USA, Cat.# M0482S) was used in the RT-PCR. The thermocycling conditions were initial denature at 94 °C for 2 min, 25 or 28 cycles of 94 °C for 30 s, 53 °C for 30 s, and 68 °C for 30 s, followed by a final extension at 68 °C for 5 min. The 10 ul of PCR products were loaded to the 1% agarose (Fisher Scientific, USA, Cat.# BP1356-500) gel, and the GeneRuler 1 kb DNA Ladder (Thermo Scientific, USA, Cat.# SM0314) was used as a molecular marker.

**Quantification of gene expression in transgenic plants by qRT-PCR**. Leaf tissue was sampled at 12 days after inoculation for RNA extraction using the RNeasy Plant Mini Kit (Qiagen, Germany, Cat.# 74904). Verso cDNA Synthesis Kit (Thermo Scientific, USA, Cat.# AB1453A) was used for cDNA Synthesis. The cDNA was input in 10 ul reaction for quantitative PCR (qPCR) using IQ SYBR Green Supermix (Bio-rad, USA, Cat.# 1708882) on the CFX96 Touch Real-Time PCR Detection System (Bio-rad, USA). Primers Lr42-qRT_F6 and Lr42-qRT_R6 (Supplementary Data 8) were used for *Lr42*, and primers actin_F1 and actin_R1 (Supplementary Data 8) were used for the *actin* gene as the control. The thermocycling conditions were initial denature at 95 °C for 3 min, 40 cycles of 95 °C for 10 s, 60 °C for 30 s. The ΔΔCT method was used to determine the relative expression of *Lr42*[28,65].

**Quantification of genomic copy number of the *Lr42* transgene**. Genomic DNAs of single plants were extracted from leaf tissue using DNeasy Plant Mini Kit (Qiagen, Germany) and quantified using Qubit 1X dsDNA High Sensitivity Assay kit (Invitrogen, USA, Cat.# Q33230). Total 10 ng genomic DNA was input for qPCR using IQ SYBR Green Supermix (Bio-rad, USA, Cat.# 1708882) on the CFX96 Touch Real-Time PCR Detection System(Bio-rad, USA) following the manufacturer's instructions. Primers Lr42_difBW_3F and Lr42-qRT_R6 were used for *Lr42*, and primers actin_F1 and actin_R1 were used for the *actin* gene (Supplementary Data 8). Similar to the analysis of qRT-PCR, the *actin* gene was used as the control to determine the DNA level of the *Lr42* transgene in transgenic lines and other control lines, including the Thatcher line that carries *Lr42* (Thatcher-*Lr42*)[77]. The *Lr42* DNA level of each line was then normalized to the *Lr42* copy number relative to Thatcher-*Lr42*, which was considered to carry one copy of *Lr42* in the 3x wheat genome because the *Lr42* was introduced by crossing with an *Lr42* line and maintained as the *Lr42* homozygous line.

**Conserved domain and repeats annotation**. Protein and DNA sequences were submitted to NCBI for conserved domain search[78]. LRR was searched by a web-based LRR search tool with additional manual examination[79].

**Development of *Lr42* diagnostic markers on the *Lr42* gene**. Multiple alignment of *Lr42* alleles from the *Ae. tauschii* minicore set identified a unique region (~140 bp) in the LRR (*Lr42*-unique-segment) from the *Lr42* resistant allele. We attempted to design diagnostic markers on the *Lr42* gene across this region. First, the *Lr42* sequence of the unique region was aligned to all *Lr42* homologs in the reference genomes of *Ae. tauschii*, wild emmer, durum wheat, CS, Barley, and Brachypodium. The top hit was a homolog (1D:7381846-7384626) in CS. The top hit sequence carries two SNPs with the *Lr42* unique sequence. Outside this highly similar region, high polymorphisms were found between *Lr42* and the homolog 1D:7381846-7384626. The sequence of the second best hit has 19 SNPs, confirming that the *Lr42*-unique-segment is not common in diverse genomes. Based on this finding, for each KASP assay, we designed a *Lr42* specific primer on the *Lr42*-unique-segment, and a primer on a homolog from the cluster of *Lr42* homologs on CS 1D. The common primer paired with them was designed on a conserved region between *Lr42* and the homolog. The primer pair that amplifies the *Lr42* homolog could potentially amplify a paralog in *Ae. tauschii* genomes. Therefore, in most populations, the assay is considered to be a dominant marker for detection of the *Lr42* resistant allele.

**Reporting summary**. Further information on research design is available in the Nature Research Reporting Summary linked to this article.

## Data availability
The BSR-Seq sequencing data generated in this study have been deposited in the Sequence Read Archive (SRA) database under accession PRJNA604114, and Nanopore whole genome sequencing data of TA2450 under accession PRJNA769399. The sequence of the *Lr42* resistance allele was deposited in GenBank under accession OK430880. Source data are provided with this paper.

## Code availability
Related scripts are available at GitHub (https://github.com/PlantG3/Lr42) or Zenodo (https://zenodo.org/badge/latestdoi/414748290).

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

## Acknowledgements

Funding was provided by the Kansas Wheat Commission award (no. KWC2020-08) to S.L.; the NSF award (no. 1238313) to B.G.; and the NSF award (no. 1822162) to J.P. Additional funding to support this project includes the USDA NIFA awards (nos. 2018-67013-28511 and 2021-67013-35724) to S.L., the NSF awards (nos. 1741090 and 2011500) to S.L., the NSF award (no. 1943155) to N.R. and V.K.T., and the USDA NIFA awards (nos. 2020-67013-32558 and 2020-67013-31460) to N.R. and V.K.T. We thank Dr. Christopher Toomajian for the discussion, Dr. Alina Akhunova from the Integrative Genomics Facility at Kansas State University for preparation of RNA-Seq libraries, John Raupp for maintaining and sharing *Ae. tauschii* accessions, Shuangye Wu for DNA extraction and genotyping, Dr. Hongliang Wang for *Lr42* genotyping of wheat populations, Andrew Fan Bai for peeling *Ae. tauschii* seeds, and Taylor Schulden for rust inoculation and phenotyping. This is contribution 20-131-J from the Kansas Agricultural Experiment Station, Manhattan, Kansas.

## Author contributions

B.G. and S.L. conceptualized experiments; G.L., H.C., and S.K.S. conducted molecular experiments; G.L., S.K.S., D.L.W., H.S., and R.L.B. developed mapping population and performed disease phenotyping; B.T., H.L., and H.N.T. performed transformation; L.S., N.R., V.K.T., and A.W.S. performed the VIGS experiment and helped mutant screening; M.J.G. and G.B. validated KASP markers; G.L., J.X., P.J., N.R., V.K.T., N.S., S.S., J.P.,

R.L.B., and S.L. analyzed data; P.J., R.P.S., and J.P. provided genotyping and phenotyping data of CIMMYT lines; G.L., R.L.B., G.B., B.G., and S.L. wrote the manuscript; all authors reviewed and revised the manuscript.

## Competing interests

S.L. is the co-founder of Data2Bio, LLC. Other authors claim no competing interest.
