## [Peer Review File · Nature Communications]

Cloning of the Broadly Effective Wheat Leaf Rust Resistance Gene Lr42 Transferred from *Aegilops tauschii*Reviewers' Comments:

Reviewer #1:

Remarks to the Author:

This manuscript reported cloning of leaf rust resistance gene Lr42 present in the wheat wild relative *Aegilops tauschii* accession TA2450 using a BSR-Seq mapping strategy. The authors identified an NLR ORF as the Lr42 candidate and transformation of the ORF driven by the maize ubiquitin promoter Ubi-1 in susceptible cultivar Bobwhite confirmed the association.

The following are my comments:

1. The Lr42 candidate was deduced based on the *Ae. tauschii* reference genome that carries the susceptible allele. It would be conclusive if a physical sequence map of the Lr42 donor was made available. The BSR-Seq mapping approach could only identify reads that map to the reference genome. False positives would occur if large sequence variations exist in the Lr42 target interval. It is well known this is very likely for R genes.
2. The association of the NLR ORF was confirmed by the ubiquitin promoter Ubi-1-driven ORF transformation. This could also be misleading. A typical example was the cloning of wheat powdery mildew resistance gene Pm21 (PNAS (2011) 108, 7727-7732 <https://doi.org/10.1073/pnas.1016981108>), which was later demonstrated to be wrong (Molecular Plant, 2018, <https://doi.org/10.1016/j.molp.2018.03.004>, <https://doi.org/10.1016/j.molp.2018.02.013>). I would suggest transformation with the native promoter and a mutant analysis or gene-editing for the candidate gene confirmation.
3. NLR type R genes are always race-specific. Now that Lr42 showed effectiveness against all leaf rust races so far, what is the underlying mechanism as to the resistance conferred by the cloned NLR candidate?
4. Line 180-184, What role is played by the ~140 bp unique Lr42 segment in the LRR domain in leaf rust resistance? A non-allelic Lr42 homolog carrying a similar segment with 98% identity was identified in Chinese Spring. Was there any functional implication?
5. Line 114 & Supplementary Figure 1, The segregation ratio in the TA2450 x TA10132 population might indicate an additional gene in the population functioned together with Lr42 in conferring the resistance.
6. In Line 144 & Supplementary Table 3, what were the six *P. triticina* races screened? The Supplementary Table 3 is hard to understand for general readers without an explanation.
7. Line 165-170. The language is really hard to follow. Please revise.
8. Line 206-232 could be put in the Supplementary file.
9. Line 146, is Supplementary Figure 7 correct there?
10. Line 302, are all the lines in Supplementary Table 12 possess Lr42?
11. Line 163, should supplementary Table 4 be cited after '3 L1 accessions'? Moreover, why didn't the authors provide the leaf rust infection type from PNMRJ infection for all the lines? An expected band does not mean a functional Lr42 allele. Sequence comparisons and their association with the infection types are important.
12. Are the sequences of Lr42 and its susceptible alleles available in the public domain such as the NCBI database?

13. Conventionally, Lr42 indicates the resistant allele, with lr42 indicating the susceptible allele. In this paper, the resistant type has also been designated as Lr42R. Please revise to maintain consistency and follow the norm.

14. In general, the contents and explanation of all the Supplementary Tables were not straightforward to the context of the main text.

Reviewer #2:

Remarks to the Author:

Paper by Lin et al. describes the cloning of a valuable, widely used in breeding programs, wheat rust resistance gene Lr42 from *Aegilops tauschii*. The main result, cloning of the functional gene, is well described and the authors presented sufficient proof that they cloned the correct gene. The authors also carried additional work to investigate homologs of the functional gene in close relatives, which is a nice addition to a cloning paper. Overall, the experiments are well designed and the results presented in a clear, easy to follow way.

The paper overall is written fine, but it is not always clear what the main message is of the paragraph. The paper would benefit from a couple of sentences of introduction for each section.

I don't have any major comments, there are several specific points in the order of the text:

- Line 137; authors refer to Figure 2C to comment on polymorphisms between resistant and susceptible alleles. However, the figure presents the protein structures of the Lr42R and lr42S alleles, which is somehow confusing;
- Line 155; 'We amplified the Lr42 homologs from each of 35 out of 40' this statement is technically not correct. Authors attempted to amplify the sequence from 35 accessions but managed to do it only from 11, please rephrase;
- Authors speculate about the origin of Lr42 only based on its uniqueness and phylogenetic analysis. However, authors did a great job to define orthologues from additional *A. tauschii* accessions, wheat and also *Brachypodium*. It would be interesting to run some analysis to see what evolutionary forces are driving this locus (I believe dataset should be sufficient to try dN/dS, Tajima's D or McDonald Kreitmann test);
- Line 195; there is a reference to Figure 3C, shouldn't it be 3D? The same, I believe, is true for line 267. The text doesn't correctly refer to Figure 3C with polymorphism sliding window analysis?
- Line 290; explanation that resistance in transgenic lines with strong ubiquitin promoter is stronger than in WT hexaploidy line is highly plausible. I'm wondering, however, whether the authors attempted transformation with native promoter? I believe it is a good practice to use native promoters for complementation analysis. Could authors please comment on this?
- In supplementary figures where alignments are presented, in the captions, authors provide information that alignments were generated using Geneious; Geneious per se is not performing alignments, it's using ClustalW as stated in Material and Methods section, please rephrase.

Reviewer #3:

Remarks to the Author:

In this study, Lin et al. report the cloning of the wheat leaf rust resistance gene Lr42. This gene has been introgressed into the hexaploid bread wheat gene pool from the diploid wild wheat progenitor *Aegilops tauschii* and has subsequently been bred into many wheat cultivars. The authors used bulked segregant RNA-Seq and genetic mapping to delimit the Lr42 gene to a 116 kb interval that contained three candidate genes. They show through gene transformation that a nucleotide binding – leucine-

rich repeat (NLR) gene is Lr42.

Generally, the paper is well written, although the section describing the haplotype analysis could be shortened (lines 155-204). Although gene cloning in wheat has become simpler, the cloning of a wheat leaf rust resistance gene still is a major accomplishment. However, I have two major concerns regarding the candidate gene identification and validation.

1) The candidate gene identification (presented in Figure 1d) was done based on the genome sequence of the *Ae. tauschii* reference accession AL8/78. As far as I understand from this manuscript, AL8/78 does not have the Lr42 resistance gene. It is known that NLRs often show extensive copy number variation between different genotypes. It is thus possible that the Lr42-carrying accession Ta2450 might have additional copies of this NLR and the authors provide no information about the length and gene content of the target interval in a Lr42-carrying genotype.

2) In line 52, the authors state that 'a nucleotide-binding site leucine-rich repeat (NLR) gene was sufficient to confer strong resistance to leaf rust'. The authors decided to use a cDNA construct driven by maize ubiquitin promoter for gene validation. I strongly question this decision. There has been recent evidence that methods based on gene overexpression might have led to misinterpretations in the field of plant-pathogen interactions. In the case of Lr42, the different phenotypic responses in diploid and polyploid wheat provide a strong indication that expression levels might play an important role in regulating the resistance response. In my view, the authors need to provide additional evidence that they cloned the correct gene. This evidence needs to be based on a genomic construct driven by its native promoter and/or a complete knock-out (mutagenesis / CRISPR) of the gene.

Minor comments:

- Line 78: I suggest using the term 'all-stage resistance' instead of 'both seeding and adult stage resistance'.
- Line 79: 'The Lr42 gene is effective against all reported races of the leaf rust fungus in the US'. Please be more specific. The cited reference specifically looked at *P. triticina* isolates collected during 2017. Has virulence on Lr42 been reported? The authors provide some explanation on this in the discussion, but I suggest to move this to the introduction
- Lines 125-127: '...that flank three other genes...'. Is this based on the *Ae. tauschii* reference genome? If so, is the reference accession a Lr42 carrier? If not, it needs to be stated that the physical interval might look different in TA2450.
- Line 164: Should it be 'from the *Ae. tauschii* reference genome AND of the leaf rust susceptible...'? Or is the reference accession AL8/78 = TA 10132?
- Line 79: The authors should sequence the complete Lr42 gene from some key hexaploid wheat cultivars like Century and Thatcher Lr42. This is important to demonstrate that the authors cloned the correct gene.

Reviewers' comments:

Reviewer #1 (Remarks to the Author):

This manuscript reported cloning of leaf rust resistance gene *Lr42* present in the wheat wild relative *Aegilops tauschii* accession TA2450 using a BSR-Seq mapping strategy. The authors identified an NLR ORF as the *Lr42* candidate and transformation of the ORF driven by the maize ubiquitin promoter *Ubi-1* in susceptible cultivar Bobwhite confirmed the association.

The following are my comments:

[R1-Q1]: 1. The *Lr42* candidate was deduced based on the *Ae. tauschii* reference genome that carries the susceptible allele. It would be conclusive if a physical sequence map of the *Lr42* donor was made available. The BSR-Seq mapping approach could only identify reads that map to the reference genome. False positives would occur if large sequence variations exist in the *Lr42* target interval. It is well known this is very likely for *R* genes.

[R1-Q1 Response]: The reviewer is correct that it would be valuable to figure out the physical map of the resistant haplotype of the *Lr42* locus, which likely possesses duplication, deletion, and/or sequence rearrangement. Following the suggestion, we first identified the relatively conserved sequence regions based on the CGRD (comparative genome read depth) analysis using publicly available whole genome shotgun (WGS) Illumina sequences from both the reference line and TA2450 (Letter Figure 1a). Second, we produced >10x WGS Nanopore long-read sequencing data for a local assembly of the *Lr42* interval, resulting in the *Lr42* locus of 201,365 bp. The comparison between the *Lr42* locus and the reference homologous locus showed that no local duplication of *Lr42* in the *Lr42* locus. The *Lr42* gene is located at a relatively conserved region, but its promoter is divergent from the reference (Letter Figure 1b).

Letter Figure 1. Sequence comparison of the *Lr42* locus with the reference.

(a). Read depth comparison through CGRD. Y-axis indicates the similarity between two genomes, green lines represent conserved regions and blue lines represent polymorphic or deleted regions in TA2450 relative to the reference genome (TA10132). (b) The sequence comparison between the newly assembled *Lr42* locus and the reference genome. The red and orange rectangles signify the location of the *Lr42* gene and the *Lr42* homologs with a relatively low identity on the *Lr42* locus, respectively. Blue rectangles signify all annotated genes in the reference sequence.

[R1-Q2]: 2. The association of the NLR ORF was confirmed by the ubiquitin promoter *Ubi-1*-driven ORF transformation. This could also be mis-leading. A typical example was the cloning of wheat powdery mildew resistance gene *Pm21* (PNAS (2011) 108, 7727-7732 <https://doi.org/10.1073/pnas.1016981108>), which was later demonstrated to be wrong (Molecular Plant,

2018, <https://doi.org/10.1016/j.molp.2018.03.004>, <https://doi.org/10.1016/j.molp.2018.02.013>). I would suggest transformation with the native promoter and a mutant analysis or gene-editing for the candidate gene confirmation.

[Our response]: To provide additional evidence, as suggested, we performed three additional experiments: an *Lr42* transformation experiment with the native promoter and the native terminator (exp1), a knockdown experiment using VIGS (exp2), and screening TA2450 EMS mutants for *Lr42* mutants (exp3). From exp1, two independent *Lr42* transgenic lines using susceptible Bobwhite were obtained. Both transgenic lines displayed resistance. The resistance levels were found to be correlated with expression levels of *Lr42*. In exp2, the VIGS experiment was performed using TA2450, the parental resistant line carrying *Lr42*, to suppress *Lr42* expression levels. We observed pustules (spores) in the VIGS plants with the *Lr42* construct but not in the VIGS plants with the empty construct. In addition, by screening 1,320 M3 families of EMS mutants derived from TA2450, we found one family showing the resistant and susceptible segregation. The amino acid substitution caused by an EMS-type single nucleotide mutation was identified in the LRR region of *Lr42*. Genotyping of this mutation of individuals in the mutant family showed co-segregation of phenotypes and genotypes. Our efforts have strengthened the conclusion that the NLR gene we identified is the *Lr42* gene. We have added all these results to the revised manuscript.

[R1-Q3]: 3. NLR type R genes are always race-specific. Now that *Lr42* showed effectiveness against all leaf rust races so far, what is the underlying mechanism as to the resistance conferred by the cloned NLR candidate?

[Our response]: We provided a thought that “*Lr42* is currently deployed mainly in wheat lines from CIMMYT that contain combinations of durable adult plant resistance (APR) genes to leaf rust. This may have reduced the selection pressure on the pathogen population to overcome *Lr42*.” in the original Discussion. We have added another explanation in the Discussion: “It is possible that the effector gene conferring *Lr42* resistance is important for the fungus, which could explain why no virulent rust isolates have been identified.”.

[R1-Q4]: 4. Line 180-184, What role is played by the ~140 bp unique *Lr42* segment in the LRR domain in leaf rust resistance? A non-allelic *Lr42* homolog carrying a similar segment with 98% identity was identified in Chinese Spring. Was there any functional implication?

[Our response]: This is a good question. So far, we do not know the role of this 140-bp segment. With more wheat genome assemblies available, the potential function of this segment can be explored in the near future.

[R1-Q5]: 5. Line 114 & Supplementary Figure 1, The segregation ratio in the TA2450 x TA10132 population might indicate an additional gene in the population functioned together with *Lr42* in conferring the resistance.

[Our response]: The reviewer is correct that the homozygote of susceptible plants is underrepresented. It implies the involvement of multiple resistance genes or the segregation distortion due to the linkage of *Lr42* with some other genes associated with the fitness. The importance is that we clearly map the resistance to a single locus, which is concordant with the mapping locus from the other mapping location.

[R1-Q6]: 6. In Line 144 & Supplementary Table 3, what were the six *P. triticina* races screened? The Supplementary Table 3 is hard to understand for general readers without an explanation.

[Our response]: Thank you for pointing this out. We have corrected six to three. The list in the **Supplementary Table 3** refers to the leaf rust resistance genes. We have added “Lr” to each ID to make it clearer.

Race	Avirulent	Virulent
PNMRJ	Lr2a, Lr11, Lr14a, Lr16, Lr17, Lr21, Lr26, Lr42	Lr1, Lr2c, Lr3, Lr3ka, Lr9, Lr10, Lr18, Lr24, Lr28, Lr30, Lr39, LrB
TFBJG	Lr3ka, Lr9, Lr11, Lr16, Lr17, Lr18, Lr21, Lr30, Lr39, Lr42, LrB	Lr1, Lr2a, Lr2c, Lr3, Lr10, Lr14a, Lr24, Lr26, Lr28
MBJ/SP	Lr2a, Lr2b, Lr2c, Lr3ka, Lr9, Lr16, Lr18, Lr19, Lr21, Lr24, Lr25, (Lr26), Lr28, Lr29, Lr30, Lr32, Lr33, Lr36, Lr42	Lr1, Lr3, Lr3bg, Lr10, Lr11, Lr13, Lr15, Lr17, Lr20, Lr23, Lr27+Lr31

[R1-Q7]: 7. Line 165-170. The language is really hard to follow. Please revise.

[Our response]: The paragraph has been revised: “Of all TA10132 *Lr42* homologs, the homolog with the highest similarity to *Lr42* and located in the *Lr42* mapping interval is deemed to be the allelic homolog of *Lr42* (*lr42-TA10132* or *lr42*). The *lr42-TA10132* allele was used in the transgenic experiment. Among all *Ae. tauschii* *Lr42* homologs, 10 homologs amplified from 10 *Ae. tauschii* minicore accessions are most similar to *Lr42*, supporting that these 10 homologs are also allelic to *Lr42* (Fig. 4a).”

[R1-Q8]: 8. Line 206-232 could be put in the Supplementary file.

[Our response]: Thank you for the suggestion. These two paragraphs attempt to tie the *Lr42* gene to the breeding applications. The paragraph of “*Lr42* in the CIMMYT programs” is particularly important to understand how *Lr42* has been utilized and how effective the resistance of *Lr42* is in breeding programs. We think it would be valuable to keep both paragraphs in the main text.

[R1-Q9]: 9. Line 146, is Supplementary Figure 7 correct there?

[Our response]: We assume that the reviewer referred to Line 246, the Supplementary Figure 7 is a phylogenetic tree of known NLR proteins conferring rust resistance. It is the correct citation.

[R1-Q10]: 10. Line 302, are all the lines in Supplementary Table 12 possess *Lr42*?

[Our response]: We attempted to genotype these lines, but we failed to secure the materials for genotyping. Based on the pedigree information, all these 11 lines possibly carry *Lr42*. Among them, Koshan 09, Gambo and Ekinoks have the same pedigree as the QUAIU line. In the Supplementary Table 10 (sample order 63), the line derived from QUAIU #1 carries *Lr42*.

[R1-Q11]: 11. Line 163, should supplementary Table 4 be cited after “3 L1 accessions”? Moreover, why didn’t the authors provide the leaf rust infection type from PNMRJ infection for all the lines? An expected band does not mean a functional *Lr42* allele. Sequence comparisons and their association with the infection types are important.

[Our response]: Thank the reviewer for pointing this out. We have moved the citation of Supplementary Table 4 after “3 L1 accessions”. We fully agree with the reviewer that expected band does not mean a functional *Lr42* allele. As suggested, we have collected phenotypic data of infection type from PNMRJ. We have revised the result based on the new infection data.

[R1-Q12]: 12. Are the sequences of *Lr42* and its susceptible alleles available in the public domain such as the NCBI database?

[Our response]: The sequence of the *Lr42* allele including the promoter and terminator regions has been deposited to GenBank. The GenBank accession number is OK430880. Other alleles from *Ae. tauschii* can be found in Source data of Figure 3a (file: sourcedata.figure3a.fasta).

[R1-Q13]: 13. Conventionally, *Lr42* indicates the resistant allele, with *lr42* indicating the susceptible allele. In this paper, the resistant type has also been designated as *Lr42R*. Please revise to maintain consistency and follow the norm.

[Our response]: Thank you for pointing this out. We have replaced all *Lr42R* to *Lr42*, and *lr42S* with *lr42*.

[R1-Q14]: 14. In general, the contents and explanation of all the Supplementary Tables were not straight-forward to the context of the main text.

[Our response]: We have gone through each Supplement Table and tried our best to improve their connection with the main text.

=====

Reviewer #2 (Remarks to the Author):

Paper by Lin et al. describes the cloning of a valuable, widely used in breeding programs, wheat rust resistance gene *Lr42* from *Aegilops tauschii*. The main result, cloning of the functional gene, is well described and the authors presented sufficient proof that they cloned the correct gene. The authors also carried additional work to investigate homologs of the functional gene in close relatives, which is a nice addition to a cloning paper. Overall, the experiments are well designed and the results presented in a clear, easy to follow way.

[R2-Q1]: The paper overall is written fine, but it is not always clear what the main message is of the paragraph. The paper would benefit from a couple of sentences of introduction for each section.

[Our response]: We have added a brief introduction for each section in Results. Thank the reviewer for the suggestion.

I don't have any major comments, there are several specific points in the order of the text:

[R2-Q2]: Line 137; authors refer to Figure 2C to comment on polymorphisms between resistant and susceptible alleles. However, the figure presents the protein structures of the *Lr42R* and *lr42S* alleles, which is somehow confusing;

[our response]: Thank you very much for pointing this out. The Supplementary Figure 4 is the correct figure to cite. We have corrected it.

[R2-Q3]: Line 155; 'We amplified the *Lr42* homologs from each of 35 out of 40' this statement is technically not correct. Authors attempted to amplify the sequence from 35 accessions but managed to do it only from 11, please rephrase;

[Our response]: We have rephrased the statement: "We examined the *Lr42* homologs from 35 minicore accessions, ...". Thanks.

[R2-Q4]: Authors speculate about the origin of *Lr42* only based on its uniqueness and phylogenetic analysis. However, authors did a great job to define orthologues from additional *A. tauschii* accessions, wheat and also *Brachypodium*. It would be interesting to run some analysis to see what evolutionary forces are driving this locus (I believe dataset should be sufficient to try *dN/dS*, Tajima's *D* or McDonald Kreitmann test);

[Our response]: We thank the reviewer for the suggestion. We agree with the reviewer that the evolution of NLR genes is an interesting topic. However, based on our data, we have difficulty to figure out a reasonable hypothesis for this analysis. We calculated the *dN/dS* and Tajima's *D* values and performed McDonald Kreitmann test using the 12 *Lr42* allelic homologs of *A. tauschii* as well as using 53 *Lr42* homologs from *A. tauschii* accessions and closed related

species (e.g., wheat, *Brachypodium*). From the results (**Letter Table 1**) we could not draw a solid conclusion without controls. We thank the reviewer for the specific methodology guidance. We would appreciate additional guidance if the reviewer views this evolutionary analysis as an important addition to the manuscript.

Letter Table 1. Selection of the *Lr42* homologs

Group	dN/dS	Tajima's D
12 allelic homologs	0.464	0.627
53 homologs	0.449	-1.329

McDonald–Kreitman test
Neutrality index = 1.03
Fisher's exact test p=0.88

[R2-Q5]: Line 195; there is a reference to Figure 3C, shouldn't it be 3D? The same, I believe, is true for line 267. The text doesn't correctly refer to Figure 3C with polymorphism sliding window analysis?

[Our response]: Again, thank you very much. We have corrected them.

[R2-Q6]: Line 290; explanation that resistance in transgenic lines with strong ubiquitin promoter is stronger than in WT hexaploidy line is highly plausible. I'm wondering, however, whether the authors attempted transformation with native promoter? I believe it is a good practice to use native promoters for complementation analysis. Could authors please comment on this?

[Our response]: During the revision, we have performed a new *Lr42* transformation experiment with the native promoter and the native terminator. As a result, two independent *Lr42* transgenic lines using susceptible Bobwhite were obtained. Both transgenic lines displayed resistance. We have added this new result to the revised manuscript.

[R2-Q7]: In supplementary figures where alignments are presented, in the captions, authors provide information that alignments were generated using Geneious; Geneious per se is not performing alignments, it's using ClustalW as stated in Material and Methods section, please rephrase.

[Our response]: Thanks for pointing this out. We have added ClustalW to the Methods.

=====

Reviewer #3 (Remarks to the Author):

In this study, Lin et al. report the cloning of the wheat leaf rust resistance gene *Lr42*. This gene has been introgressed into the hexaploid bread wheat gene pool from the diploid wild wheat progenitor *Aegilops tauschii* and has subsequently been bred into many wheat cultivars. The authors used bulked segregant RNA-Seq and genetic mapping to delimit the *Lr42* gene to a 116 kb interval that contained three candidate genes. They show through gene transformation that a nucleotide binding – leucine-rich repeat (NLR) gene is *Lr42*.

[R3-Q1]: Generally, the paper is well written, although the section describing the haplotype analysis could be shortened (lines 155-204).

[our response]: Thank you for pointing this out. We have revised the paragraph to clarify the description.

Although gene cloning in wheat has become simpler, the cloning of a wheat leaf rust resistance gene still is a major accomplishment. However, I have two major concerns regarding the candidate gene identification and validation.

[R3-Q2]: 1) *The candidate gene identification (presented in Figure 1d) was done based on the genome sequence of the Ae. tauschii reference accession AL8/78. As far as I understand from this manuscript, AL8/78 does not have the Lr42 resistance gene. It is known that NLRs often show extensive copy number variation between different genotypes. It is thus possible that the Lr42-carrying accession Ta2450 might have additional copies of this NLR and the authors provide no information about the length and gene content of the target interval in a Lr42-carrying genotype.*

[Our response]: The reviewer is correct that the NLR region frequently contains duplication, deletion, and/or sequence rearrangement among genomes. In the revised manuscript, we produced >10x whole genome shotgun (WGS) Nanopore long-read sequencing data and performed a local assembly of the *Lr42* interval, which resulted in the *Lr42* locus of 201,365 bp. The comparison between the *Lr42* locus and the reference homologous locus showed that no local duplication of *Lr42* in the *Lr42* locus.

[R3-Q3]: 2) *In line 52, the authors state that ‘a nucleotide-binding site leucine-rich repeat (NLR) gene was sufficient to confer strong resistance to leaf rust’. The authors decided to use a cDNA construct driven by maize ubiquitin promoter for gene validation. I strongly question this decision. There has been recent evidence that methods based on gene overexpression might have led to misinterpretations in the field of plant-pathogen interactions. In the case of Lr42, the different phenotypic responses in diploid and polyploid wheat provide a strong indication that expression levels might play an important role in regulating the resistance response. In my view, the authors need to provide additional evidence that they cloned the correct gene. This evidence needs to be based on a genomic construct driven by its native promoter and/or a complete knock-out (mutagenesis / CRISPR) of the gene.*

[Our Response]: As suggested by the reviewer, we have performed a new *Lr42* transformation experiment with the native promoter and the native terminator and screening TA2450 EMS mutants for *Lr42* mutants. We have identified two independent *Lr42* transgenic lines using susceptible Bobwhite. Both transgenic lines displayed resistance. The resistance levels appear to be correlated with expression levels of *Lr42*. We also screened 1320 M3 families of EMS mutants derived from TA2450, identifying one mutant family that had segregating resistant and susceptible individuals. Sequencing the EMS mutant identified an amino acid substitution caused by an EMS-type single nucleotide mutation in the LRR region of *Lr42*. Genotyping of this mutation on individuals of the family showed co-segregation of phenotypes and genotypes. We have added these results to the revised manuscript.

Minor comments:

[R3-Q4]: *Line 78: I suggest using the term ‘all-stage resistance’ instead of ‘both seeding and adult stage resistance’.*

[Our response]: This has been modified. Thanks.

[R3-Q5]: *Line 79: ‘The Lr42 gene is effective against all reported races of the leaf rust fungus in the US’. Please be more specific. The cited reference specifically looked at P. triticina isolates collected during 2017. Has virulence on Lr42 been reported? The authors provide some explanation on this in the discussion, but I suggest to move this to the introduction*

[Our response]: To date, no virulence has been identified. The reviewer is correct that the cited reference in the original manuscript only includes isolates from 2017. In the updated studies that collected more recent isolates until 2020, there are still no virulent isolates identified (references below). We therefore had the statement that “The *Lr42* gene is effective against all reported races”. In the revised manuscript, we have added more references to support this statement.

1. Kolmer, J.A. and Fajolu, O.L. 2021. Wheat leaf rust in the United States in 2020. *Wheat Newsletter*. 67:78-87
2. Kolmer, J.A. and Fajolu, O.L. 2020. Wheat leaf rust in the United States in 2019. *Wheat Newsletter*. 66:89-91

[R3-Q6]: Lines 125-127: ‘...that flank three other genes...’. Is this based on the *Ae. tauschii* reference genome? If so, is the reference accession a *Lr42* carrier? If not, it needs to be stated that the physical interval might look different in TA2450.

[Our response]: The statement is based on the reference genome. We have sequenced and assembled the *Lr42* locus using Nanopore reads. The sequence comparison showed both conservation and divergence in the locus. The result has been added to the revised manuscript.

[R3-Q7]: Line 164: Should it be ‘from the *Ae. tauschii* reference genome AND of the leaf rust susceptible...’? Or is the reference accession AL8/78 = TA 10132?

[Our response]: Yes, AL8/78=TA10132. TA10132 is the susceptible line used to construct the *Ae. tauschii* reference genome. To clarify the description, the revision has been made as: “We also extracted intact *Lr42* homologs from the TA10132, the *Ae. tauschii* accession for the reference genome. TA10132, also known as AL8/78, is a leaf rust susceptible accession.”.

[R3-Q8]: Line 79: The authors should sequence the complete *Lr42* gene from some key hexaploid wheat cultivars like Century and Thatcher *Lr42*. This is important to demonstrate that the authors cloned the correct gene.

[Our response]: The *Lr42* CDS from the hexaploid wheat cultivars KS91WGRC11 and KS93U50 have been amplified and sequenced. The sequence data showed 100% identity to the *Lr42* from its donor line TA2450. No *Lr42* homologs can be amplified from the susceptible line Century. For Thatcher *Lr42*, we did amplify *Lr42* fragments containing multiple NLR homologs but failed to specifically amplify *Lr42* for sequencing.

Reviewers' Comments:

Reviewer #1:

Remarks to the Author:

In this revision, the authors added investigation of the sequence structure of the Lr42 interval and provided a Lr42 transformation experiment using the native promoter, a VIGS experiment, and a mutant analysis for Lr42 mutant analysis, which is essential, besides other revisions. It is a pity that as an additional member of the NLR family, the mechanism of Lr42 on Lr resistance was not addressed. There are still other concerns too.

1. Now that the Lr42 interval sequence has been assembled, the authors should state directly that how many genes exist in the TA2450 sequence interval, how many of them are different from the susceptible parent in the regulatory regions or CDS regions in terms of nucleotide composition, instead of focusing on the reference genome annotations. The section from line 221 to line 341 should be placed immediately after the physical mapping part. The current structure is really awkward. Is there a possibility that genes other than AET1Gv20040300 are the Lr42 candidates? For instance, are the two additional Lr42 homologs located in the 116 kb interval? Are they expressed? If they are within the interval, there are at least five genes in the TA2450 interval. One could not simply assume that they are not candidates because the sequences are different from AET1Gv20040300.
2. The criteria for disease score seems varied in the research. For example in Fig. 2b, the Lr42p::Lr42-2 were scored as -2, while those, such as TA10141 and TA2448, had a similar phenotype (Supp Table 4, Supp Fig. 11) were scored as 3 (susceptible). In Supplementary Fig. 7a, was the disease symptom sufficiently developed? It was quite different from other presentations of this paper. These inconsistencies will compromise the conclusion.
3. Line 183-186, I assume that the three genes in the 116 kb interval flanked by pC43 and pC50 were those annotated in the reference genome. Please state clearly the annotated genes in TA2450.
4. Fig. 3c, expression of Lr42 in TA2450 should be included as a control.
5. Lr42 has been reported as a recessive gene by the same group? (Liu et al. 2013, Crop Sci 53:1566-1570), which is inconsistent with the findings of this study. Please explain.
6. Please indicate how many biological replicates were used in the VIGS and expression experiments?
7. In Fig. 2c, Lr42 has fewer LRR repeats compared with Lr42. Please provide detail of structural difference, which is useful for readers. Are all the proteins encoded by the Lr42 alleles in Supplementary Table 4 and Supplementary Fig. 4a were similarly shortened?
8. Fig. 1, please indicate genetic distance and physical position of the markers in Fig. 1d.

Reviewer #2:

Remarks to the Author:

The authors sufficiently addressed all my comments and added the required data. In my opinion, the paper is now suitable for publication in Nature Communications.

Reviewer #3:

Remarks to the Author:

In this revised version, the authors addressed all my queries. The new transgenic lines, the VIGS and the EMS mutant demonstrate that the candidate NLR gene confers the leaf resistance. In the response letter, the authors state that "The Lr42 CDS from the hexaploid wheat cultivars KS91WGRC11 and KS93U50 have been amplified and sequenced. The sequence data showed 100% identity to the Lr42 from its donor line TA2450." I did not find this information in the main text (maybe I missed it). This important information should be included in the manuscript because it shows that the NLR cloned from *Ae. tauschii* is Lr42.

Below are a few minor comments that the authors might address:

Lines 63-66: In the meanwhile, two additional leaf rust resistance genes have been cloned, Lr14a and Lr13. Please update the list here.

Line 69: The following reference providing an actual example of a wheat stem rust resistance gene cassette should be mentioned here: Luo et al. (2021) Nature Biotechnology 39: 561-566

Line 94: Why do you use the phrase 'ectopic' expression? This would indicate an abnormal gene expression.

Line 150: A reference to the 'publically available' whole genome sequencing data is missing.

Lines 158 – 161: These sentences here are a bit confusing. What do you mean by 60% coverage? Is there clear evidence that the two homologous copies are truncated? Do you mean that 'indicating that the Lr42 locus only contains a single full-length copy of Lr42'?

Lines 186 – 188: Please provide exact numbers here. How many individuals from the respective M3 family were used?

#####

Reviewer 1

#####

In this revision, the authors added investigation of the sequence structure of the Lr42 interval and provided a Lr42 transformation experiment using the native promoter, a VIGS experiment, and a mutant analysis for Lr42 mutant analysis, which is essential, besides other revisions.

[Our response]: Thank you for the reviewer to agree that we have provided essential data for the revision.

It is a pity that as an additional member of the NLR family, the mechanism of Lr42 on Lr resistance was not addressed. There are still other concerns too.

[Our response]: We share the curiosity to understand how Lr42 works. In particular, we would like to figure out why Lr42 is still highly effective. The cloning of the resistance gene is a key step for further exploration. For example, our work will facilitate cloning of the effector gene from the pathogen and studying pathogen-host interactions. We also would like to mention that the cloning of Lr42 has important positive impacts on wheat breeding. We have received requests of the Lr42 gene sequence for resistance gene stacking and breeding programs have used breeding markers we developed. We believe this study represents an important achievement to combat the wheat rust disease.

1. Now that the Lr42 interval sequence has been assembled, the authors should state directly that how many genes exist in the TA2450 sequence interval, how many of them are different from the susceptible parent in the regulatory regions or CDS regions in terms of nucleotide composition, instead of focusing on the reference genome annotations. The section from line 221 to line 341 should be placed immediately after the physical mapping part. The current structure is really awkward. Is there a possibility that genes other than AET1Gv20040300 are the Lr42 candidates? For instance, are the two additional Lr42 homologs located in the 116 kb interval? Are they expressed? If they are within the interval, there are at least five genes in the TA2450 interval. One could not simply assume that they are not candidates because the sequences are different from AET1Gv20040300.

[Our response]: Thank the reviewer for suggestions. We have provided more details about genes on the Lr42 interval and reorganized paragraphs. Briefly, we used the sequence information provided by the newly Lr42 assembly and clarified that two homologous NLR sequences in the locus are fragmented. In more detail, one homolog was annotated as a gene (AET1Gv20040500) containing CC-NB domains. This gene was expressed in both resistant (R) and susceptible (S) lines. However, no DNA polymorphisms were identified. The other NLR fragment was not expressed in either R and S lines and was not annotated as a gene. We therefore concluded that only one intact NLR gene is in the Lr42 locus. In the revised manuscript, this part was placed after the ubiquitin transformation experiment because we used the Lr42 gene sequence to fish Nanopore reads for the assembly.

We agree with the reviewer that homology data are not sufficient to conclude that Lr42 gene is only one gene conferring resistance. However, in our study, we showed that

the introduction of the single gene (AET1Gv20040300) into a susceptible cultivar induced resistance and knockout of the gene disrupted rust resistance, confirming that the candidate gene is required and sufficient for the *Lr42*-mediated resistance. Therefore, our results strongly support that AET1Gv20040300 is the gene conferring the *Lr42* resistant phenotype.

2. *The criteria for disease score seems varied in the research. For example in Fig. 2b, the Lr42p::Lr42-2 were scored as -2, while those, such as TA10141 and TA2448, which had a similar phenotype (Supp Table 4, Supp Fig. 11) were scored as 3 (susceptible). In Supplementary Fig. 7a, was the disease symptom sufficiently developed? It was quite different from other presentations of this paper. These inconsistencies will compromise the conclusion.*

[Our response]: Thank the reviewer for the careful review. We tried our best to consistently score phenotypes among different experiments. We employed a standard phenotyping approach and used the controls included in each experiment as references. Host responses to pathogens varied among individual plants and leaves. We assigned a summary phenotyping score for each line based on predominant phenotypes. Based on the reviewer's feedback, we re-evaluated and adjusted phenotype scores to 3- for both TA10141 and TA2448. Thanks. The disease symptom from the VIGS experiment was indeed fully developed. The intermediate phenotype in the VIGS experiment was consistent with the partial knockdown of gene expression of *Lr42* as documented in Supplementary Fig. 7b. This is also consistent with the results in Fig. 3b and 3c that show an excellent correlation between resistance level and *Lr42* gene expression level. The *Lr42* knockdown experiment was conducted from the University of Maryland. Both groups from Kansas State University and University of Maryland shared a common inoculation protocol. However, the effective inoculum dose in the Maryland experiment was lower than Kansas experiments for unknown reasons and that resulted in fewer pustules per leaf. Nevertheless, the infection types were clear and consistent in Supplementary Fig. 7a.

3. *Line 183-186, I assume that the three genes in the 116 kb interval flanked by pC43 and pC50 were those annotated in the reference genome. Please state clearly the annotated genes in TA2450.*

[Our response]: We have added more details in the sequence comparison between TA2450 and the reference genome (the new figure Fig. 3b). Homologs of all three genes can be found in the *Lr42* locus and they are collinear.

4. *Fig. 3c, expression of Lr42 in TA2450 should be included as a control.*

[Our response]: We shared the reviewer's thoughts when we designed the experiment. However, we realized that TA2450 and wheat lines are from two species. We need to use different control genes. And copy numbers of the control genes in diploid and hexaploid lines are different. We, therefore, think the quantitative qRT-PCR results from two species are not comparable.

5. *Lr42* has been reported as a recessive gene by the same group? (Liu et al. 2013, *Crop Sci* 53:1566-1570), which is inconsistent with the findings of this study. Please explain.

[Our response]: [Our response]: The reviewer is correct that *Lr42* was reported as the recessive resistance allele in Liu et al 2013. Our manuscript and other studies (Sun et al. 2010; Cox, Raupp, and Gill 1994; Gill et al. 2019) showed that the *Lr42* resistance allele is dominant. Kolmer and Dyck (1994) demonstrated that, depending on the genotypes of the host lines and pathogen isolates used, expression of resistance genes could range from dominant to recessive. In our manuscript, the diploid *Ae. tauschii* population was used for mapping because the gene is very strong and effective in the diploid source. On the other hand, Liu et al 2013 used a hexaploid wheat population where the gene provides only intermediate resistance. The discrepancy of allelic dominance effects probably results from the influence of *Lr42* gene expression in different genetic backgrounds.

Cox, T. S., W. J. Raupp, and B. S. Gill. 1994. "Leaf Rust-Resistance Genes Lr41, Lr42, and Lr43 Transferred from *Triticum Tauschii* to Common Wheat." *Crop Science* 34. Madison, WI: Crop Science Society of America: 339–43.

Gill, Harsimardeep S., Chunxin Li, Jagdeep S. Sidhu, Wenxuan Liu, Duane Wilson, Guihua Bai, Bikram S. Gill, and Sunish K. Sehgal. 2019. "Fine Mapping of the Wheat Leaf Rust Resistance Gene Lr42." *International Journal of Molecular Sciences* 20 (10). doi:10.3390/ijms20102445.

Kolmer, J. A., and P. L. Dyck. 1994. "Gene Expression in the *Triticum Aestivum*-*Puccinia Recondita* F. Sp. *Tritici* Gene-for-Gene System." *Phytopathology* 84 (4). [St. Paul, Minn., etc.: American Phytopathological Society]: 437–40.

Liu, Zhengli, Robert L. Bowden, and Guihua Bai. 2013. "Molecular Markers for Leaf Rust Resistance Gene Lr42 in Wheat." *Crop Science* 53. Madison, WI: The Crop Science Society of America, Inc.: 1566–70.

Sun, Xiaochun, Guihua Bai, Brett F. Carver, and Robert Bowden. 2010. "Molecular Mapping of Wheat Leaf Rust Resistance Gene Lr42." *Crop Science* 50. Madison, WI: Crop Science Society of America: 59–66.

6. Please indicate how many biological replicates were used in the VIGS and expression experiments?

[Our response]: Thank the reviewer for pointing this out. Three biological replicates were used. We have added this to the Methods.

7. In Fig. 2c, *lr42* has fewer LRR repeats compared with *Lr42*. Please provide detail of structural difference, which is useful for readers. Are all the proteins encoded by the *lr42* alleles in Supplementary Table 4 and Supplementary Fig. 4a were similarly shortened?

[Our response]: The detailed structural difference is now added in Supplementary Fig. 5. The main difference started from the 10th LRR repeat. Of 11 *lr42* alleles, eight encode truncated proteins, ranging from 116 to 840 amino acids in length. Other three proteins have a similar length to *Lr42*, which has 920 amino acids.

Supplementary Fig. 5. Protein alignment of Lr42 and Ir42-TA10132. The conserved amino acid residues between two sequences were indicated with black boxes. Fourteen LRR repeats in Lr42 and 11 LRR repeats in Ir42-TA10132 were identified. The LRR repeats were labeled with dark gray arrows.

8. Fig. 1, please indicate genetic distance and physical position of the markers in Fig. 1d.
[Our response]: Physical positions of all markers are listed in the Supplemental Table 1. We have labeled positions of some markers in Fig. 1d. We did not specifically determine genetic distance among markers.

 Reviewer 2

 The authors sufficiently addressed all my comments and added the required data. In my opinion, the paper is now suitable for publication in Nature Communications.

 Reviewer 3

 In this revised version, the authors addressed all my queries. The new transgenic lines, the VIGS and the EMS mutant demonstrate that the candidate NLR gene confers the leaf resistance.

In the response letter, the authors state that “The Lr42 CDS from the hexaploid wheat cultivars KS91WGRC11 and KS93U50 have been amplified and sequenced. The sequence data showed 100% identity to the Lr42 from its donor line TA2450.”. I did not find this information in the main text (maybe I missed it). This important information should be included in the manuscript because it shows that the NLR cloned from *Ae. tauschii* is Lr42.

[Our response]: Good suggestion. We have added the result and a new supplementary figure (Supplementary Fig. 12) to the manuscript to indicate that early developed *Lr42* lines do carry the resistance allele.

Supplementary Fig. 12. Wheat accessions KS91WGRC11 and KS93U50 carry *Lr42*. *Lr42* from the wheat line KS91WGRC11 and KS93U50 was amplified and Sanger sequenced. The multiple alignment was performed using ClustalW in the Geneious. No vertical lines in three gray rectangles indicate the identity of the three sequences.

Below are a few minor comments that the authors might address:

Lines 63-66: In the meanwhile, two additional leaf rust resistance genes have been cloned, Lr14a and Lr13. Please update the list here.

[Our response]: Thank the reviewer for pointing this out. We have added these two studies in the Introduction.

Line 69: The following reference providing an actual example of a wheat stem rust resistance gene cassette should be mentioned here: Luo et al. (2021) Nature Biotechnology 39: 561-566

[Our response]: Thanks. We have added this citation.

Line 94: Why do you use the phrase ‘ectopic’ expression? This would indicate an abnormal gene expression.

[Our response]: “ectopic” rather than “overexpression” was used to indicate the gene was not expressed from the original chromosomal location. It avoids having the statement of “overexpression” without evidence.

Line 150: A reference to the ‘publically available’ whole genome sequencing data is missing.

[Our response]: Thank you! We cited the paper in the Methods and now the citation is in the Results as well.

Lines 158 – 161: These sentences here are a bit confusing. What do you mean by 60% coverage? Is there clear evidence that the two homologous copies are truncated? Do you mean that ‘indicating that the Lr42 locus only contains a single full-length copy of Lr42’?

[Our response]: We have rephrased the description. The reviewer is correct that we meant that those sequences are NLR fragments. The conclusion of our new paragraph “Long-read local assembly shows only one intact NLR in the *Lr42* locus” is that “The sequence of the *Lr42* locus shows that the locus only contains a single intact NLR gene.”.

Lines 186 – 188: Please provide exact numbers here. How many individuals from the respective M3 family were used?

[Our response]: We added the number to the sentence: “Genotyping of all individuals (N=13) in the family showed a co-segregation among genotypes and phenotypes, i.e., all homozygous mutants were susceptible and all others were resistant.”

Reviewers' Comments:

Reviewer #1:

Remarks to the Author:

The authors addressed most of my comments in this revision. But I still have two concerns critical to the conclusion.

1. In Fig. 4a, the phenotype of the Lr42p::Lr42-2 were scored as 2- (resistant) although was similar to that of a few lines such as TA2433 and TA1665, which were scored as 3 or 3+ (susceptible) in Supp Table 4, Supp Fig. 11. I have raised this concern in my previous comments.

2. The expression of Lr42 in Lr42p::Lr42-1 showed a great variation range (0.9-11) (Figure 4b, Sourcedata Figure 4b). Particularly, the expression value of Lr42p::Lr42-1 in the third replicate was 0.935, very similar to the expression of Lr42p::Lr42-2 that was susceptible in my opinion. Which of the three Lr42p::Lr42-1 replicates truly reflected the Lr42 expression in the line? Is that possible that factors other than the of transgenes lead to disease resistance in Lr42p::Lr42-1?

Reviewer #1 (Remarks to the Author):

The authors addressed most of my comments in this revision. But I still have two concerns critical to the conclusion.

1. In Fig. 4a, the phenotype of the *Lr42p::Lr42-2* were scored as 2- (resistant) although was similar to that of a few lines such as TA2433 and TA1665, which were scored as 3 or 3+ (susceptible) in Supp Table 4, Supp Fig. 11. I have raised this concern in my previous comments.

[Our response]: We thank the reviewer for the very careful review. We apologize that we did not adequately address this question in our previous response. The reviewer's question addresses our ability to discern a resistant reaction from a susceptible reaction among the wild types, transgenics, and mutants. This ability is crucial for our conclusions to be valid. Since the reviewer has raised this point twice, we will go into some wheat rust pathology detail to show that we can indeed discern the differences.

We used the Stakman seedling infection type rating scale, which is the accepted standard around the world for wheat rust pathologists (Kolmer, J. A 1996, 2019, Roelfs et al., 1992). See Table R3-1 from Roelfs et al 1992 for the scale description. We often see variation in pustule size or associated necrosis or chlorosis between leaves and even within one leaf. The Stakman scale is capable of recording variation within a leaf in several ways. For example, infection types can be combined (e.g., 13 showing a range from 1 to 3 in pustule size), or modified with annotations for extra chlorosis (C), necrosis (N), or larger (+), or smaller (-), or much smaller (= read as minus minus) than the type. If there are discrete pustule sizes on the same leaf, the infection types can be separated by a comma (e.g., 1,3). If there is a difference among plants in an accession, the infection types can be separated by a slash mark (e.g., 1C/3C-). A range of pustule types on the same leaf can also be indicated with an "X" for heterogeneous infection type. If heterogeneous and there are larger pustules at the base, it is indicated with a "Z". If larger at the tip, it is indicated with a "Y". A large number of ratings are possible, which may be very confusing to a non-specialist (e.g. 0;, 1-, 2C, 2=, ;2+, 1+N, X+, etc.).

It is also possible to take a simplified or summary approach and limit the scale to 0, 1-, 1, 1+, 2-, 2, 2+, 3-, 3, 3+, or 4. It omits the nuances of variation, but is far easier to understand. It clarifies the essential information about resistance reaction. It also happens to be much better for machine-coding (Zhang et al, 2014). We used the Stakman summary approach in this paper, given that the audience is not just wheat rust pathologists.

In Fig. 4a (copied below in Fig. R3-1 for this response), the pustule infection type of the *Lr42p::Lr42-2* transgenic line varies from a large hypersensitive fleck with no sporulation (IT = ; which is read as "fleck") to small-sized pustules with a chlorotic halo (IT = 1) to medium-sized pustules with a chlorotic halo (IT = 2). The third leaf

has two or three medium-large-sized pustules that each could be rated IT = 2+ or even 3-. The most common individual infection type for the first two leaves is a fleck and ranges up to IT = 2. The rating for those leaves could be written as ;2 using the full Stakman scale. The flecks are less common in the third leaf and IT = 2 seems to be most common. The rating for leaf 3 could be written as ;3- given that fleck is more common than 3-. The leaves for this line, including more than just those shown in photos, were summary-rated in real time as IT = 2-, indicating that the most common infection type was a 2, but all the hypersensitive flecks and frequent IT = 1 pustules showed more resistance than a typical 2. The proper control for these transgenic plants is Bobwhite. It showed no hypersensitive flecks and a consistent 3 type pustule. There is no doubt that *Lr42p::Lr42-2* is showing moderate resistance compared to Bobwhite. *Lr42p::Lr42-2* is more susceptible than *Lr42p::Lr42-1*, but it is still definitely not in the category of “susceptible” like Bobwhite. The cut-off for “susceptible” is considered IT ≥ 3 by users of the Stakman scale.

What about lines TA2433 and TA1665? These are lines of diploid *Ae. tauschii* and inoculated with race PNMRJ. Therefore, hexaploid Bobwhite/TFBJG is not the proper control. In fact, TA2433 is the susceptible mapping parent control. See Fig. R3-2 from Fig. 1a where the pustules on TA2433 are medium to large with no flecks and very little chlorosis. In Fig. R3-1, if you look at the size of the pustules for TA2433 in relation to the leaf width, they appear to be medium-sized. Notably there are no flecks or small resistant pustules. The photo of replicate 1 for TA1665 shows nearly the same reaction as TA2433 near the top. In the middle and bottom of the leaf, the pustules appear to be smaller, but there are no hypersensitive flecks nor necrotic halos. Perhaps this photo deserves a rating of 3- rather than 3. The rating of each line was given in real time to a group of multiple leaves. Photos from replicate 2 (not shown in the manuscript but added here in proof) of independent leaves confirmed the susceptible rating results for both TA2433 and TA1665. From both replicates, neither TA2433 nor TA1665 matches *Lr42p::Lr42-2* for reaction.

In summary, we can't rule out that an occasional rust rating was slightly high or low in Supplementary Table 4. Errors are always possible. However, we do claim to have very good experience, understanding, and proficiency at rating rust resistance phenotypes. The pictures in Fig. R3-1 and our discussion above clearly show that *Lr42p::Lr42-1* is highly resistant and *Lr42p::Lr42-2* is moderately resistant compared to the Bobwhite control. This is consistent with the gene expression data.

Table R3-1. Phenotyping guidance from Roelfs et al. 1992

Table 21. Host response and infection type descriptions used in wheat stem and leaf rust systems.

Host response (class)	Infection type ^a	Disease symptoms
Immune	0	No uredinia or other macroscopic sign of infection
Nearly immune	;	No uredinia, but hypersensitive necrotic or chlorotic flecks present
Very resistant	1	Small uredinia surrounded by necrosis
Moderately resistant	2	Small to medium uredinia often surrounded by chlorosis or necrosis; green island may be surrounded by chlorotic or necrotic border
Heterogeneous	X	Random distribution of variable-sized uredinia on single leaf
Heterogeneous	Y	Ordered distribution of variable-sized uredinia, with larger uredinia at leaf tip
Heterogeneous	Z	Ordered distribution of variable-sized uredinia, with larger uredinia at leaf base
Moderately susceptible	3	Medium-sized uredinia that may be associated with chlorosis
Susceptible	4	Large uredinia without chlorosis

^a The infection types are often refined by modifying characters as follows: =, uredinia at lower size limit for the infection type; -, uredinia somewhat smaller than normal for the infection type; +, uredinia somewhat larger than normal for the infection type; ++, uredinia at the upper size limit for the infection type; C, more chlorosis than normal for the infection type; and N, more necrosis than normal for the infection type. Discrete infection types on a single leaf when infected with a single biotype are separated by a comma (e.g., 4,; or 2=, 2+ or 1,3C). A range of variation between infection types is recorded by indicating the range, with the most prevalent infection type listed first (e.g., 23 or ;1C or 31N); after Roelfs (297).

Figure R3-1. Disease phenotypes of multiple lines

Wheat lines Bobwhite and *Lr42p::Lr42-2* were phenotyped using race TFBJG. Both photos are in Fig. 4. Other *Ae. tauschii* lines were phenotyped with race PNMRJ. Note that photos are not on exactly the same magnification. *Ae. tauschii* photos of replicate 1 are in Supplemental Fig. 11. Photos of replicate 2 were added here for the reviewing purpose. The summary scoring results based on plant phenotypes are: Bobwhite: 3, *Lr42p::Lr42-2*: 2-, TA2433: 3, TA2450: ;, TA1665: 3, TA1667: 3+, TA1605: 3+.

TA2433

TA2450

Figure R3-2. Rust phenotypes of TA2433 and TA2450

Phenotypes of *Ae. tauschii* accessions TA2433 was susceptible (Infection Type = 33+) and TA2450 was hypersensitive fleck (Infection Type = ; to ;1-) at the seedling stage upon inoculation with race PNMRJ. Photos are from Fig. 1a.

References

1. Roelfs, A.P., R.P. Singh, and E.E. Saari. Rust Diseases of Wheat: Concepts and methods of disease management. Mexico, D.F.:CIMMYT. P81 (1992)
2. Kolmer, J. A. Genetics of resistance to wheat leaf rust. *Annu. Rev. Phytopathol.* 34, 435–455 (1996).
3. Kolmer, J. A. Virulence of *Puccinia triticina*, the Wheat Leaf Rust Fungus, in the United States in 2017. *Plant Dis.* 103, 2113–2120 (2019).
4. Zhang, D., Bowden, R. L., Yu, J., Carver, B. F. & Bai, G. Association analysis of stem rust resistance in U.S. winter wheat. *PLoS One* 9, e103747 (2014).

2. The expression of *Lr42* in *Lr42p::Lr42-1* showed a great variation range (0.9-11) (Figure 4b, Sourcedata Figure 4b). Particularly, the expression value of *Lr42p::Lr42-1* in the third replicate was 0.935, very similar to the expression of *Lr42p::Lr42-2* that was susceptible in my opinion. Which of the three *Lr42p::Lr42-1* replicates truly reflected the

Lr42 expression in the line? Is that possible that factors other than the of transgenes lead to disease resistance in *Lr42_p::Lr42-1*?

[Our response]: As we discussed above, *Lr42_p::Lr42-2* was scored as a moderately resistant line. The moderate resistance of *Lr42_p::Lr42-2* is presumably due to low expression of *Lr42*. The reviewer is correct that expression variation of *Lr42* among *Lr42_p::Lr42-1* individuals was large. We do not know exactly what caused that. Individuals of *Lr42_p::Lr42-1* are T1 plants. There might be multiple insertion sites of the transgene *Lr42* in the parental T0 line. Different *Lr42* insertion sites may be associated with distinct levels of *Lr42* expression. Copy number variation of *Lr42* in T1 plants due to Mendelian segregation of these insertion sites could result in variation in gene expression. In the legend of Fig. 4, we have added this note: “High variation of *Lr42* expression in *Ubi_p::Lr42* and *Lr42_p::Lr42* transgenic lines may be due to segregation of *Lr42* transgenes.”.

Reviewers' Comments:

Reviewer #1:

Remarks to the Author:

Two comments:

1. Information on how many T2 (Ubip::Lr42) or T1 (Lr42p::Lr42) plants were phenotypically evaluated should be given in Fig. 4a legend.
2. If the expression level variation in Fig. 4b was caused by segregation of Lr42p::Lr42 copy number, then how many plants were sampled for each biological replicate? The authors should quantitate the Lr42p::Lr42 copy number of each T1 plant, which is quite easy. If the Lr42 expression level was related to resistance level, the author should examine the association of the expression level of T1 transgenic plants with their resistance levels. The authors should address these questions clearly in the text or fig legend.

Reviewer #1 (Remarks to the Author):

1. Information on how many T₂ (*Ubi*::*Lr42*) or T₁ (*Lr42p*::*Lr42*) plants were phenotypically evaluated should be given in Fig. 4a legend.

[Our response]: We thank the reviewer for the suggestion. In the revised legend, we have added numbers of transgenic lines that were phenotyped. The updated legend was included below. A small number of seeds were germinated due to the limited available T₁ seeds. Below is the updated legend of Figure 4. Information added was highlighted in yellow.

Fig. 4: Ectopic *Lr42* expression and the EMS *Lr42* mutant. (a) Phenotype of Bobwhite was susceptible (N=12, Infection Type=3), phenotype of *Ubi*::*Lr42* (*Lr42* driven by the maize ubiquitin promoter) T₂ transgenic plants (event 3) was hypersensitive fleck (N=12, Infection Type=;), and phenotype of two independent events of *Lr42p*::*Lr42* (*Lr42* driven by the native promoter of *Lr42*) were resistant upon inoculation with race TFBJG. Infection types of *Lr42p*::*Lr42-1* (N=3) and *Lr42p*::*Lr42-2* (N=9) were ; and 2-, respectively. Transgenic plants of both *Lr42p*::*Lr42* events were in T₁ generation, and *Ubi*::*Lr42* plants were from T₂ generation. (b) qRT-PCR of the *Lr42* expression in Bobwhite and transgenic plants. *Lr42* expression of Bobwhite is undetectable. One plant was used for each biological replicate in gene expression analysis. (c) Phenotype of M3 individuals from the TA2450 EMS mutant family that carried a G to A mutation at 2099 bp of the *Lr42* coding region, causing the C700Y amino acid substitution. The genotype (GG, GA, or AA) at the mutation site of each plant individual was listed above the leaf. Seedling plants were inoculated with race PNMRJ. Source data are provided as a Source Data file.

2. If the expression level variation in Fig. 4b was caused by segregation of *Lr42p*::*Lr42* copy number, then how many plants were sampled for each biological replicate? The authors should quantitate the *Lr42p*::*Lr42* copy number of each T₁ plant, which is quite easy. If the *Lr42* expression level was related to resistance level, the author should examine the association of the expression level of T₁ transgenic plants with their resistance levels. The authors should address these questions clearly in the text or fig legend.

[Our response]: For gene expression analysis, single plants were examined. As the reviewer suggested, we have performed qPCR to quantify the *Lr42* copy number for single plants with genomic DNAs available, including samples from three *Ubi*::*Lr42-3* plants, three *Lr42p*::*Lr42-1*, and five *Lr42p*::*Lr42-2*. The Thatcher carrying *Lr42* (Thatcher-*Lr42*), Thatcher (no *Lr42*), and Bobwhite (no *Lr42*) were used as the control. Presumably Thatcher-*Lr42* carries one copy of *Lr42* in the 3x wheat genome because the

Lr42 was introduced by crossing with an *Lr42* line and maintained as the *Lr42* homozygous line. As compared to *Lr42_p::Lr42-2*, *Ubi_p::Lr42-3* and *Lr42_p::Lr42-1* exhibited higher *Lr42* expression and stronger resistance. However, copy number analysis (Fig. R4-1) indicated that *Lr42_p::Lr42-2* T1 plants carried higher copies of *Lr42*. In addition, copy number variation presumably due to segregation was observed among individuals of *Ubi_p::Lr42-3* and *Lr42_p::Lr42-2*. The result indicated that *Lr42* insertion copies in transgenic lines differ in transcription and influence on rust resistance, which is likely due to the position effect of the transgene and/or gene silencing caused by a high copy of transgenes (Allen et al. 1996, Feng et al. 2001).

Expression of the transgene *Lr42* and the resistance level have been directly compared (Fig. R4-2), indicative of a non-linear relationship between them. It appears that a high level of rust resistance requires a certain level of *Lr42* expression, and, at low levels of expression, resistance increases with the elevation of *Lr42* expression.

In the revised manuscript, we have integrated these results. We thank the reviewer for pointing this out. It is valuable to document the possible position effect of the transgene. And we hope our effort has clarified the association between *Lr42* expression and resistance.

Figure R4-1. Genomic copy number estimation of *Lr42* via quantitative PCR. The *Lr42* copy of Thatcher-*Lr42* was normalized to 1. The copy number relative to Thatcher-*Lr42* in each other plant was quantified. Arrow bars stand for standard deviations from two technical replicates for each line. As expected, no copy was detected in either Thatcher or Bobwhite. Copy number variation was evidenced among individuals within the events of *Ubi_p::Lr42-3* and *Lr42_p::Lr42-2*. The infection type (IT) of each individual plant was labeled.

Figure R4-2. Disease resistance versus *Lr42* expression. Original seedling infection types were converted to a 0–9 disease scale (Methods). Relative expression data of *Lr42* were the same as qRT-PCR data used in Fig. 4b. Expression data were transformed with the logarithm of base 2 (log2).

References

- Allen, G. C. et al. High-level transgene expression in plant cells: effects of a strong scaffold attachment region from tobacco. *Plant Cell* 8, 899–913 (1996).
- Feng, Y. Q. et al. Position effects are influenced by the orientation of a transgene with respect to flanking chromatin. *Mol. Cell. Biol.* 21, 298–309 (2001).

Reviewers' Comments:

Reviewer #1:

Remarks to the Author:

I am not satisfied with the authors' response and revision related to my comments, since it raised more questions instead of solving the problems. Since the experiment is directly related to the Lr42 function and how the gene works, the questions should be clarified.

1. In Figure 4b and Sourcedata Figure 4b, the expression value for biological replicate 1 (Lr42p::Lr42-1-1) was 11.149 and that for biological replicate 3 (Lr42p::Lr42-1-3) was 0.935, while the copy number of Lr42p::Lr42 in both replicates was 2 as shown in Supplementary Fig. 7. Therefore, the copy number variation did not show any relation to their expression difference (Supplementary Fig.8), which is in contrast to the assumption proposed by the authors in the previous response. Moreover, in my view the Lr42p::Lr42-1 progenies should have a similar expression level, particularly in those plants with the same copy number, unless the authors could provide a solid explanation of the great difference.

2. The authors also proposed that certain level of Lr42 expression is required for a high-level resistance (L186-189). The expression level of Lr42 in Lr42p::Lr42-1-3 and Lr42p::Lr42-2-3 was similar (0.935, and 0.701) (Sourcedata Figure 4b), but their resistance had a scale of 1 and 5, respectively (Supplementary Fig.8). Was the expression difference statistically different?

Since only one plant was included in each biological replicate, an analysis of the T2 progenies is necessary.

REVIEWERS' COMMENTS

Reviewer #1 (Remarks to the Author):

I am not satisfied with the authors' response and revision related to my comments, since it raised more questions instead of solving the problems. Since the experiment is directly related to the *Lr42* function and how the gene works, the questions should be clarified.

1. In Figure 4b and Source data Figure 4b, the expression value for biological replicate 1 (*Lr42p::Lr42-1-1*) was 11.149 and that for biological replicate 3 (*Lr42p::Lr42-1-3*) was 0.935, while the copy number of *Lr42p::Lr42* in both replicates was 2 as shown in Supplementary Fig. 7. Therefore, the copy number variation did not show any relation to their expression difference (Supplementary Fig.8), which is in contrast to the assumption proposed by the authors in the previous response. Moreover, in my view the *Lr42p::Lr42-1* progenies should have a similar expression level, particularly in those plants with the same copy number, unless the authors could provide a solid explanation of the great difference.

[Our Response]

We, with a repeated experiment, have confirmed that the *Lr42* expression level of *Lr42p::Lr42-1-1* is quite different from that of *Lr42p::Lr42-1-3*. We speculate that the two plants carry different combinations of *Lr42* insertions that are capable of various levels of expression. The reviewer is correct that copy number variation did not show a clear relationship with expression of *Lr42* in our result. We have mentioned that copy number could influence expression levels of *Lr42*, which we still think is a reasonable statement. However, different transgenic insertions could have various levels of expression, which may lead to a large variation in expression among T1 plants. We have detailed this in the response to the editor's comment. See above.

2. The authors also proposed that certain level of *Lr42* expression is required for a high-level resistance (L186-189). The expression level of *Lr42* in *Lr42p::Lr42-1-3* and *Lr42p::Lr42-2-3* was similar (0.935, and 0.701) (Source data Figure 4b), but their resistance had a scale of 1 and 5, respectively (Supplementary Fig.8). Was the expression difference statistically different?

Since only one plant was included in each biological replicate, an analysis of the T2 progenies is necessary.

[Our Response]

We hypothesized that a certain level of *Lr42* expression is required for leaf rust resistance based on a strong pattern we observed in Supplementary Fig. 8, reproduced below for reference. We did not attempt to claim the statistical difference based on the technical replication of single plants. Instead, we observed the relationship between disease score and expression. For such observation, individual plants are suitable to be experimental objects since we can collect data of both *Lr42* expression and rust resistance. We appreciate the careful review and time from the reviewer. However, *Lr42* insertions in T2 plants are still segregating. The proposed experiment with pooled

samples of T2 plants will not capture the information of copy number or expression of the transgene in individual plants, or their relationships with disease scores. Therefore, the proposed T2 experiment with pooled T2 individuals would likely not provide additional information to what we observed.

Supplementary Fig. 8: Disease resistance versus *Lr42* expression. Original seedling infection types were converted to a 0–9 disease scale (Methods). Relative expression data of *Lr42* were the same as qRT-PCR data used in Fig. 4b. Expression data were transformed with the logarithm of base 2 (log₂).